# Golgi pH homeostasis stabilizes the lysosomal membrane through *N*-glycosylation of membrane proteins

Yu-shin Sou[1], Junji Yamaguchi[2,3], Keisuke Masuda[1], Yasuo Uchiyama[3], Yusuke Maeda[4], Masato Koike[1]

**Protein glycosylation plays a vital role in various cellular functions, many of which occur within the Golgi apparatus. The Golgi pH regulator (GPHR) is essential for the proper functioning of the Golgi apparatus. The lysosomal membrane contains highly glycosylated membrane proteins in abundance. This study investigated the role of the Golgi luminal pH in *N*-glycosylation of lysosomal membrane proteins and the effect of this protein modification on membrane stability using *Gphr*-deficient MEFs. We showed that *Gphr* deficiency causes an imbalance in the Golgi luminal pH, resulting in abnormal protein *N*-glycosylation, indicated by a reduction in sialylated glycans and markedly reduced molecular weight of glycoproteins. Further experiments using FRAP and PLA revealed that *Gphr* deficiency prevented the trafficking dynamics and proximity condition of glycosyltransferases in the Golgi apparatus. In addition, incomplete *N*-glycosylation of lysosomal membrane proteins affected lysosomal membrane stability, as demonstrated by the increased susceptibility to lysosomal damage. Thus, this study highlights the critical role of Golgi pH regulation in controlling protein glycosylation and the impact of Golgi dysfunction on lysosomal membrane stability.**

## Introduction

The Golgi apparatus is responsible for the modification and packaging of proteins and lipids into vesicles for transport to targeted destinations, and its integrity is tightly regulated by the luminal acidic pH (Kellokumpu et al, 2002; Klumperman, 2011). Dysregulation of the Golgi luminal pH alters protein transport and glycosylation and induces morphological changes in the Golgi apparatus (Weisz, 2003). The luminal pH of the Golgi apparatus is controlled by three different ion transport systems: the vacuolar (V)-ATPase-mediated proton pump, counter-ion (CL–) transport, and proton leak channel. Mutations in the proton pump subunit *ATP6V0A2* gene and defects in the counter-ion transporter Golgi pH regulator (also called *Gpr89*, G-protein-coupled receptor 89,

hereinafter referred to as *Gphr*) gene reportedly affect the function of the Golgi apparatus (Kornak et al, 2008; Maeda et al, 2008). In addition, NHE7 and SLC4A2 have been identified as proton leak channels involved in the Golgi pH homeostasis (Nakamura et al, 2005; Khosrowabadi et al, 2021).

Asparagine-linked glycosylation (*N*-glycosylation) is one of the most common post-translational modifications of proteins and is a pH-dependent function of the Golgi apparatus. Altering luminal acidic pH with compounds has been shown to cause abnormalities in proper protein transport and glycosylation (Maeda & Kinoshita, 2010; Kellokumpu, 2019). Elevating the luminal acidic pH with ammonium chloride has been shown to cause abnormalities in *O*-glycosylation because of the mislocalization of glycosyltransferase in endosomes (Axelsson et al, 2001). Chloroquine prevents the addition of sialic acid to *N*-glycans by elevating the pH of the Golgi apparatus (Rivinoja et al, 2006). Abnormal luminal pH has been reported to influence the activity and the correct localization of glycosyltransferases (Gawlitzek et al, 2000). In addition, numerous glycosyltransferases form dimers (Uemura et al, 2006). The observed pH sensitivity of the enzyme dimer suggests that acidic luminal pH is responsible for enzyme assembly in the Golgi apparatus (Hassinen et al, 2011). These observations indicate that glycosylation is highly sensitive to alterations in the Golgi luminal pH and may result from improper localization of glycosyltransferases. In the above-mentioned studies, the chemical compounds used may have inhibited the function of acidic organelles, including lysosomes, endosomes, and the Golgi apparatus. Therefore, mutant cells with impaired Golgi acidification are thought to be useful for analyzing pH regulation of the Golgi apparatus involved in protein glycosylation.

After *N*-glycosylation in the Golgi apparatus, glycoproteins are sorted and directed to their final intracellular destinations, which include secretory vesicles, plasma membranes, and lysosomes (Klumperman, 2011). The lysosomal membrane is a single phospholipid bilayer, which contains abundant highly glycosylated membrane proteins such as lysosome-associated membrane proteins (LAMPs) and lysosomal integral membrane proteins (LIMPs). The glycosylation of LMPs plays a crucial role in their function (Saftig & Klumperman, 2009). The lysosomal membrane on

---

[1]Department of Cell Biology and Neuroscience, Juntendo University Graduate School of Medicine, Bunkyo, Japan    [2]Laboratory of Morphology and Image Analysis, Research Support Center, Juntendo University Graduate School of Medicine, Bunkyo, Japan    [3]Department of Cellular and Molecular Neuropathology, Juntendo University Graduate School of Medicine, Bunkyo, Japan    [4]Department of Molecular Virology, Research Institute for Microbial Diseases, Osaka University, Suita, Japan

Correspondence: ysodaka@juntendo.ac.jp

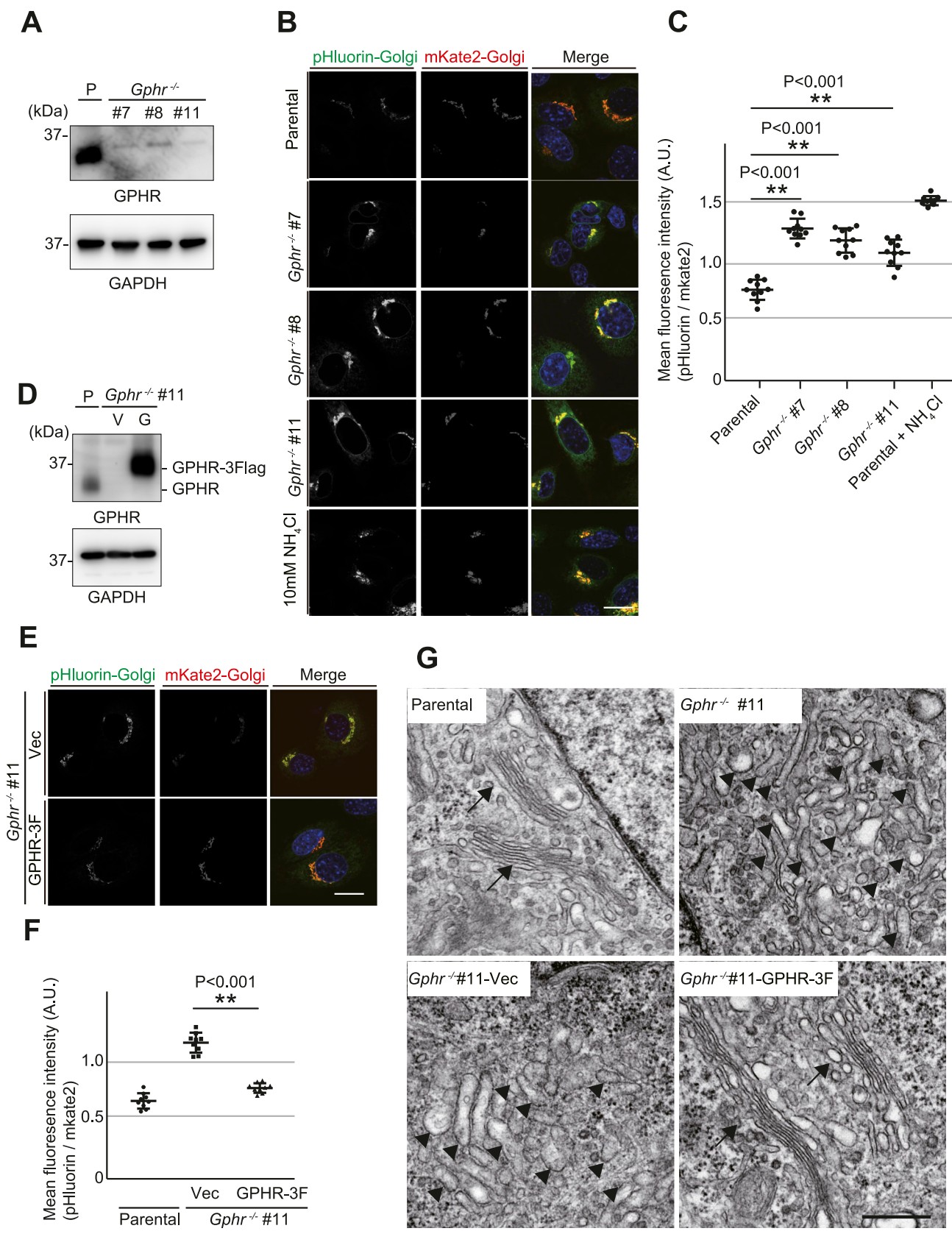

the inner side is lined with heavily glycosylated LMPs, which protect the membrane from self-degradation by lysosomal hydrolases (Saftig & Klumperman, 2009). Moreover, *N*-linked oligosaccharides of LAMP-1 and LAMP-2 have been shown to protect against degradation by lysosomal proteolysis (Kundra & Kornfeld, 1999). Along similar lines, SCAV-3, the *C. elegans* homologue of mammalian LIMP-2, maintains lysosomal integrity (Li et al, 2016). Notably, lysosomal integrity was not affected in *Lamp-1/Lamp-2* double-deficient MEFs (Eskelinen et al, 2004). Thus, the mechanism underlying lysosomal membrane integrity remains unknown. Moreover, oncogenic transformation leads to the destabilization of lysosomes because of a decrease in LAMP-1 and LAMP-2 protein levels (Fehrenbacher et al, 2008). Thus, the role of *N*-glycosylation of LMPs in maintaining lysosomal membrane integrity under physiological conditions is not fully understood.

In the present study, we aimed to investigate the physiological role of the acidic pH of the Golgi apparatus in *N*-glycosylation. When focusing on the lysosomal membrane, which contains abundant highly glycosylated proteins such as LMPs, we evaluated whether proper glycosylation contributes to membrane stability. Distinctively, our study examined the function of *N*-glycosylation in LMPs by using a *Gphr*-deficient cell model which produces incomplete *N*-type glycoproteins.

# Results

## *Gphr* is essential for proper luminal pH and morphology of the Golgi apparatus in MEFs

Previously, we have demonstrated that mutations or deficiencies in *Gphr* lead to an increase in the luminal pH of the Golgi apparatus (Maeda et al, 2008; Sou et al, 2019). To better understand the role of Golgi luminal pH in *N*-glycosylation, we used the Cre/LoxP system to generate *Gphr*-deleted MEFs derived from *Gphr* flox homozygous mice (Tarutani et al, 2012). Deletion of GPHR protein in *Gphr*-floxed MEFs expressing Cre recombinase using the adenovirus system was confirmed by immunoblot analysis using an anti-GPHR antibody (Fig 1A). As expected, luminal pH of the Golgi apparatus in *Gphr*-deficient MEFs was high, which was determined by constructing a signal localized in the Golgi apparatus for the pH-sensitive protein pHluorin (Fig 1B and C). We then expressed either vector control or GPHR-3xFlag in *Gphr*-deficient MEFs (*Gphr*$^{-/-}$ #11) and confirmed

that the luminal pH of the Golgi apparatus recovered to the same level as that of parental MEFs (Fig 1D). As shown in Fig 1E and F, the luminal pH of the Golgi apparatus was restored in *Gphr*$^{-/-}$ MEFs by the expression of GPHR-3Flag but not that of Vector control (Fig 1E and F). Ultrastructural analysis showed that *Gphr*$^{-/-}$#11 and *Gphr*$^{-/-}$ #11 expressed vector control MEFs exhibited tubulovesicular structures instead of stacks of flattened cisterns, which are characteristic of the Golgi apparatus. Abnormal morphology of the Golgi apparatus was recovered by the expression of GPHR-3xFlag (Fig 1G). These results support the concept that *Gphr* is restored in *Gphr*$^{-/-}$ #11 MEFs expressed GPHR-3xFlag (*Gphr*$^{-/-}$GPHR-3F) but not in MEFs expressed vector control (*Gphr*$^{-/-}$Vec). Hereafter *Gphr*$^{-/-}$GPHR-3F and *Gphr*$^{-/-}$Vec were used as control and *Gphr*-deficient MEFs, respectively. We also observed elevated luminal pH and abnormal morphology of the Golgi apparatus in acute knockout of *Gphr* in MEFs and *Gphr*-deficient N2A cells (Figs S1A–D and S2A–D). Taken together, we concluded that GPHR is responsible for maintaining normal luminal acidic conditions and morphology of the Golgi apparatus in cultured murine cells.

## Abnormal protein *N*-glycosylation in *Gphr*-deficient MEFs

To investigate the effect of GPHR-mediated pH regulation in the Golgi apparatus on protein *N*-glycosylation, we conducted glycomic analysis of *N*-glycans derived from total proteins using Matrix Assisted Laser Desorption Ionization Time OF Flight Mass Spectrometry (MALDI-TOF MS). The representative MALDI-TOF MS spectrum showed similar patterns of *N*-glycans in *Gphr*$^{-/-}$GPHR-3F and *Gphr*$^{-/-}$Vec MEFs (Fig 2A). The N-glycan profile showed a total of 40 *N*-glycans on fibroblasts, including high-mannose– and complex-type glycans (Table 1). Comparative analysis of *N*-glycans revealed an increase in high-mannose type *N*-glycans in *Gphr*$^{-/-}$Vec MEFs, whereas the presence of sialylated complex-type *N*-glycans decreased in *Gphr*$^{-/-}$Vec MEFs compared with that in the *Gphr*$^{-/-}$GPHR-3F (Fig 2B). The glycomic results were further confirmed by lectin blot analysis using *Sambucus nigra* (SNA) lectin as a probe to detect sialic acid in *N*-glycosylated residues. SNA reactivity was lower in the *Gphr*$^{-/-}$Vec group than in the parental and *Gphr*$^{-/-}$GPHR-3F groups (Fig 2C and D). Immunoblotting analysis revealed a markedly reduced molecular weight of CD63 (known as LIMP-1), a highly *N*-glycosylated protein, in *Gphr*$^{-/-}$Vec MEFs (Fig 2E). Targeting CD63 with peptide-*N*-glycosidase F (PNGase F), a type of *N*-glycanase, resulted in proteins of the same relative molecular

**Figure 1. Generation of Gphr-deficient MEFs.**
**(A)** Immunoblotting. Cell lysates from parental and *Gphr*-deficient MEFs (#7, #8, and #11) were subjected to immunoblotting for GPHR and GAPDH. **(B)** Fluorescence images of pHluorin-Golgi and mKate2-Golgi in parental and *Gphr*-deficient MEFs. Ammonium chloride (NH₄Cl, 10 mM), which increases the fluorescence intensity of pHluorin, was used as a positive control. Data are representative of three independent experiments. Cell nuclei were stained with Hoechst. Scale bar: 20 *μm*. **(B, C)** Quantification of pH of the Golgi apparatus. Bar graphs indicate the relative fluorescence intensity of pHluorin/mKate2 in MEFs, as described in (B). Data are means ± SD of individual values (*n* = 9 images, <300 cells examined). Significant differences and *P*-values between MEFs are indicated. *P*-values were corrected for one-way ANOVA with Tukey's *post hoc* test. **(D)** Immunoblotting. Cell lysates prepared from parental (P) and *Gphr*-deficient MEFs expressing the control vector (V) or GPHR-3xFlag (G) were subjected to immunoblotting with the indicated antibodies. **(E)** Fluorescence images of pHluorin-Golgi and mKate2-Golgi in *Gphr*$^{-/-}$Vec and *Gphr*$^{-/-}$GPHR-3F MEFs. Data are representative of three independent experiments. Cell nuclei were stained with Hoechst. Scale bar: 20 *μm*. **(F)** Quantification of pH of the Golgi apparatus. **(E)** Bar graphs indicate the relative fluorescence intensity of pHluorin/mKate2 in MEFs, as described in (E). Data are means ± SD of individual values (*n* = 9 images, <300 cells examined). Significant differences and *P*-values between MEFs are indicated. *P*-values were corrected for one-way ANOVA with Tukey's *post hoc* test. **(G)** Ultrastructural analysis of the Golgi apparatus in the parental, *Gphr*$^{-/-}$ #11, *Gphr*$^{-/-}$Vec and *Gphr*$^{-/-}$GPHR-3F MEFs. Arrows indicate the Golgi apparatus consisting of stacked flattened cisterns, and arrowheads indicate the tubulovesicular structures. Scale bar: 500 nm.
Source data are available for this figure.

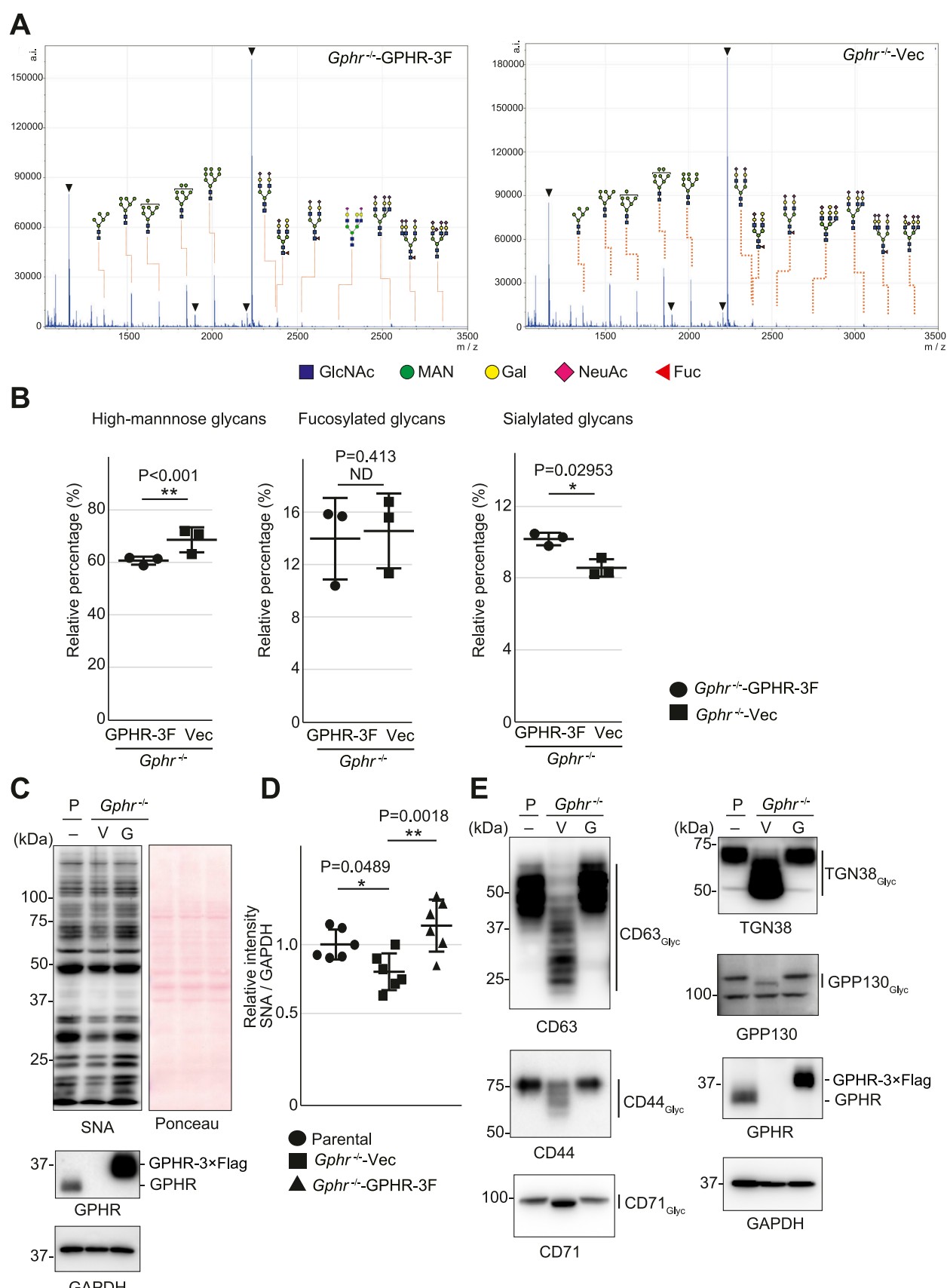

weight in parental, _Gphr_$^{-/-}$GPHR-3F, and _Gphr_$^{-/-}$Vec MEFs, indicating that _N_-glycosylation was affected (Fig S3). Furthermore, the molecular weights of other endogenous glycoproteins such as CD44, TGN38, CD71 (transferrin receptor protein 1), and GPP130 were similarly affected in _Gphr_$^{-/-}$Vec MEFs (Fig 2E). To determine whether abnormal glycosylation occurred in _Gphr_-deficient MEFs because of changes in glycosyltransferases, we performed immunoblotting for glycosyltransferases and found no significant differences in protein levels of glycosyltransferase (Fig S4A and B). We further confirmed abnormal glycosylation because of _Gphr_ deficiency by using _Gphr_$^{-/-}$ clonal MEFs, an acute _Gphr_ knockout system, and _Gphr_$^{-/-}$ clonal N2A cells (Fig S5A–C). Together, these results suggest that _N_-glycosylation is impaired as a result of _Gphr_ deficiency-induced dysregulation of the luminal pH in the Golgi apparatus.

### Luminal pH influences dynamics of glycosyltransferase trafficking in the Golgi apparatus

Glycosylation is sensitive to changes in the luminal pH of the Golgi apparatus, which may be caused by mislocalized glycosyltransferases (Axelsson et al, 2001; Rivinoja et al, 2006). To determine glycosyltransferase localization in _Gphr_-deficient MEFs, we performed immunofluorescence analysis using carboxyl-terminal GFP-fused glycosyltransferases. MEFs stably expressing either _β_-1, 4-galactosyltransferase 1-GFP (B4galt1-GFP) or _β_-galactoside _α_-2,6-sialyltransferase 1-GFP (St6gal1-GFP) were fixed and immunostained with GPP130, a marker for the Golgi apparatus. B4galt1-GFP and St6gal1-GFP showed almost exclusive localization to the Golgi apparatus in both MEFs (Fig 3A and B). Based on this observation, we hypothesized that the dynamics of glycosyltransferase trafficking in the Golgi apparatus may have been altered in _Gphr_-deficient MEFs. Numerous glycosyltransferases in the Golgi apparatus have been reported to undergo constant recycling to maintain their correct localization (Liu et al, 2018). To investigate whether _Gphr_ deficiency influences the trafficking dynamics of glycosyltransferases in the Golgi apparatus, we used FRAP assay. FRAP analysis by photobleaching an elliptical area of the Golgi apparatus (Fig 3C and Video 1, Video 2, Video 3, and Video 4) showed that _Gphr_ deficiency led to a significant delay in the recovery of fluorescent signals of B4galt1-GFP or St6gal1-GFP within the bleached area of the Golgi apparatus, compared with that in the control (Fig 3D). Furthermore, we confirmed similar results for B4galt1-GFP and St6gal1 in _Gphr_-deficient N2A cells (Fig S6A and B). These results suggested that _Gphr_ deficiency influences the trafficking dynamics of glycosyltransferases in the Golgi apparatus.

### _Gphr_ deficiency affects the proximity condition of glycosyltransferase

Several studies in eukaryotes have shown that many glycosyltransferases interact with sequentially acting enzymes within each _N_-glycosylation pathway (Kellokumpu et al, 2016). This interaction between glycosyltransferase contributes to the efficient glycosylation of substrate proteins. Co-immunoprecipitation and fluorescence resonance energy transfer have been used to analyze glycosyltransferase interaction (Hassinen & Kellokumpu, 2014; Bart et al, 2015). To determine whether the decrease in the glycosyltransferase interaction is because of the loss of trafficking dynamics in the Golgi apparatus, we performed a proximity ligation assay (PLA). PLA is a highly sensitive method of detecting protein–protein interactions, a double-antibody stain that emits a highly amplified signal only when the two probed epitopes are within 40 nm of each other (Soderberg et al, 2006). We first tested the co-localization of endogenous B4galt1 and St6gal1-GFP. MEFs stably expressing St6gal1-GFP were fixed and immunostained with B4galt1 antibody. The fluorescence image showed that the B4galt1 signal co-localized with that of St6gal1-GFP in each MEF (Fig 4A). Quantification confirmed that there was no significant difference in the co-localization rate of the B4galt1–St6gal1-GFP signal in _Gphr_$^{-/-}$Vec MEFs compared with that in _Gphr_$^{-/-}$GPHR-3F MEFs (Fig 4B). We used PLA to determine the proximity condition between B4galt1 and St6gal1. MEFs were fixed, labeled with primary antibodies, and then treated with secondary antibodies attached to DNA tags. When close enough (<40 nm), the DNA tags facilitate ligation and amplification and emit a fluorescent signal. The combination of goat B4galt1 and mouse GFP antibodies also revealed the presence of the B4galt1 and St6gal1 proximity condition (Fig 4C). The number of dots detected using these antibodies was significantly higher than the background level obtained by counting signals from the B4galt1 antibody alone (Fig 4C). The PLA signals were detected in the St6gal1-GFP signal (Fig 4C). Quantification confirmed a significant decrease in the number and area of PLA signals in _Gphr_$^{-/-}$Vec MEFs compared with those in the _Gphr_$^{-/-}$GPHR-3F MEFs (Fig 4D). To confirm that the signals observed in MEFs were indeed specific, we performed several technical controls using antibodies against Golgi markers according to their distance from B4galt1. First, we performed PLA using a combination of B4galt1 and TGN38 antibodies, which are markers of the trans-Golgi network. As expected, the number and area of PLA signals were not significantly different between the MEFs (Fig S7A–D). Second, we used Vti1a, v-SNARE localized to the trans-Golgi

---

**Figure 2. Abnormal protein glycosylation in Gphr-deficient MEFs.**
**(A)** Representative MALDI-TOF mass spectra of _N_-glycans derived from _Gphr_$^{-/-}$ GPHR-3F and _Gphr_$^{-/-}$Vec MEFs. Peaks labeled with arrowhead (▲) represent an internal standard glycan spiked into each sample. The colored symbol and nomenclature for the glycan structure follow the designation of the Consortium for Functional Glycomics. Data shown are representative of three independent experiments (_n_ = 3). **(B)** Relative quantitative comparison of high-mannose, fucosylated, and sialylated glycans. Bar graphs indicate the relative percentage of _N_-glycans in each MEF from three independent experiments (_n_ = 3). Data are means ± SD. Significant differences between MEFs are indicated (*_P_ < 0.05, **_P_ < 0.01). _P_-values were corrected using Welch's _t_ test. **(C)** SNA lectin blotting. Cell lysates prepared from parental [P], _Gphr_$^{-/-}$Vec [V] and _Gphr_$^{-/-}$GPHR-3F MEFs [G] were subjected to lectin blotting for SNA; GAPDH was used as a loading control. **(D)** Quantification of the SNA intensities. Bar graphs indicate quantitative densitometric analysis of SNA relative to GAPDH. Data are means ± SD of six separate experiments (_n_ = 6). The signal intensities of SNA and GAPDH were measured by densitometry. The value of parental MEFs was set to 1. _P_-values were corrected for one-way ANOVA with Tukey's _post hoc_ test. **(E)** Immunoblotting of glycoproteins in _Gphr_-deficient MEFs. Cell lysates were prepared from parental [P], _Gphr_$^{-/-}$Vec [V], and _Gphr_$^{-/-}$GPHR-3F MEFs [G] and then subjected to immunoblotting with the indicated antibodies.
Source data are available for this figure.

**Table 1.  Identification of N-linked glycans derived from Gphr⁻/⁻GPHR-3F and GPHR⁻/⁻Vec MEFs.**

| Glycan Code | Calc. m/z | Proposed structure | Hex | compositions NexNAc | Fuc | NeuAC | NeuGc | Gphr⁻/⁻GPHR-3F | Gphr⁻/⁻Vec |
|---|---|---|---|---|---|---|---|---|---|
| 3 2 0 0 0 | 1038.37 | | 3 | 2 | 0 | 0 | 0 | 2.76 (± 0.44) | 2.45 (± 0.74) |
| 3 2 1 0 0 | 1184.43 | | 3 | 2 | 1 | 0 | 0 | 2.03 (± 0.37) | 2.02 (± 0.50) |
| 3 3 0 0 0 | 1241.45 | | 3 | 3 | 0 | 0 | 0 | ND | 1.04 (± 0.26) |
| 6 1 0 0 0 | 1321.45 | N/A | | | | | | 2.30 (± 0.15) | 1.98 (± 0.25) |
| 5 2 0 0 0 | 1362.48 | | 5 | 2 | 0 | 0 | 0 | 5.20 (± 1.01) | 5.26 (± 1.38) |
| 3 3 1 0 0 | 1387.51 | | 3 | 3 | 1 | 0 | 0 | ND | 1.48 (± 0.16) |
| 3 4 0 0 0 | 1444.53 | | 3 | 4 | 0 | 0 | 0 | ND | 0.84 (± 0.00) |
| 7 1 0 0 0 | 1483.51 | N/A | | | | | | 1.22 (± 0.09) | 1.06 (± 0.05) |
| 6 2 0 0 0 | 1524.53 | | 6 | 2 | 0 | 0 | 0 | 10.40(± 0.52) | 10.65 (± 0.53) |
| 5 3 0 0 0 | 1565.56 | | 5 | 3 | 0 | 0 | 0 | 1.04 (± 0.02) | 0.91 (± 0.09) |
| 3 4 1 0 0 | 1590.59 | | 3 | 3 | 1 | 0 | 0 | 1.40 (± 0.04) | 2.88 (± 0.38) |
| 7 2 0 0 0 | 1686.59 | | 7 | 2 | 0 | 0 | 0 | 8.59 (± 0.15) | 9.67 (± 0.24) |
| 4 4 1 0 0 | 1752.64 | | 4 | 3 | 1 | 0 | 0 | ND | 0.89 (± 0.25) |
| 5 4 0 0 0 | 1768.64 | | 5 | 4 | 0 | 0 | 0 | ND | 1.28 (± 0.03) |
| 3 5 1 0 0 | 1793.67 | | 3 | 5 | 1 | 0 | 0 | ND | 0.99 (± 0.09) |
| 8 2 0 0 0 | 1848.64 | | 8 | 2 | 0 | 0 | 0 | 15.67 (± 0.99) | 17.70 (± 0.97) |
| 5 3 0 1 0 | 1870.67 | | 5 | 3 | 0 | 1 | 0 | 0.94 (± 0.25) | 0.81 (± 0.22) |
| 5 4 1 0 0 | 1914.70 | | 5 | 4 | 1 | 0 | 0 | 1.21 (± 0.01) | 1.52 (± 0.26) |
| 6 4 0 0 0 | 1930.69 | | 6 | 4 | 0 | 0 | 0 | ND | 0.78 (± 0.07) |
| 9 2 0 0 0 | 2010.69 | | 9 | 2 | 0 | 0 | 0 | 20.89 (± 2.41) | 15.89 (± 1.05) |
| 6 3 0 1 0 | 2032.72 | | 6 | 3 | 0 | 1 | 0 | 0.97 (± 0.00) | 0.77 (± 0.23) |
| 5 4 0 1 0 | 2073.75 | | 5 | 4 | 0 | 1 | 0 | 1.94 (± 0.60) | 1.52 (± 0.43) |
| 6 4 1 0 0 | 2076.75 | | 6 | 4 | 1 | 0 | 0 | 1.13 (± 0.13) | 1.13 (± 0.02) |
| 7 4 0 0 0 | 2092.74 | | 7 | 4 | 0 | 0 | 0 | 1.12 (± 0.00) | 1.01 (± 0.06) |
| 5 5 1 0 0 | 2117.78 | | 5 | 5 | 1 | 0 | 0 | 0.91 (± 0.02) | ND |
| 10 2 0 0 0 | 2172.74 | | 10 | 2 | 0 | 0 | 0 | 3.13 (± 0.16) | 1.89 (± 0.05) |
| 5 4 1 1 0 | 2219.81 | | 5 | 4 | 1 | 1 | 0 | 1.55 (± 0.08) | 1.67 (± 0.10) |
| 7 4 1 0 0 | 2238.80 | | 7 | 4 | 1 | 0 | 0 | 2.4 (± 0.10) | 1.85 (± 0.33) |
| 6 5 1 0 0 | 2279.83 | | 6 | 5 | 1 | 0 | 0 | 0.69 (± 0.02) | 0.57 (± 0.03) |
| 5 4 0 2 0 | 2378.86 | | 5 | 4 | 0 | 2 | 0 | 2.20 (± 0.43) | 1.92 (± 0.69) |
| 6 4 1 1 0 | 2381.86 | | 6 | 4 | 1 | 1 | 0 | 4.15 (± 0.04) | 3.1 (± 0.12) |
| 5 4 1 2 0 | 2524.92 | | 5 | 4 | 1 | 2 | 0 | 2.08 (± 0.03) | 1.87 (± 0.35) |
| 6 5 1 1 0 | 2584.94 | | 6 | 5 | 1 | 1 | 0 | 0.59 (± 0.04) | 0.45 (± 0.00) |
| 6 5 0 2 0 | 2743.99 | | 6 | 5 | 0 | 2 | 0 | 0.71 (± 0.1) | 0.68 (± 0.00) |
| 6 5 1 2 0 | 2890.05 | | 6 | 5 | 1 | 2 | 0 | 0.44 (± 0.00) | 034 (± 0.01) |
| 8 5 1 1 0 | 2909.05 | | 8 | 5 | 1 | 1 | 0 | 0.52 (± 0.02) | 0.42 (± 0.00) |
| 6 5 0 3 0 | 3049.10 | | 6 | 5 | 0 | 3 | 0 | 1.56 (± 0.17) | 1.28 (± 0.28) |
| 7 5 1 2 0 | 3052.10 | | 7 | 5 | 2 | 0 | 1 | 0.89 (± 0.05) | 0.62 (± 0.04) |
| 6 5 1 3 0 | 3195.16 | | 6 | 5 | 1 | 3 | 0 | 0.51 (± 0.00) | 0.41 (± 0.07) |
| 6 5 0 4 0 | 3354.22 | | 6 | 5 | 0 | 4 | 0 | 0.41 (± 0.07) | 0.34 (± 0.09) |

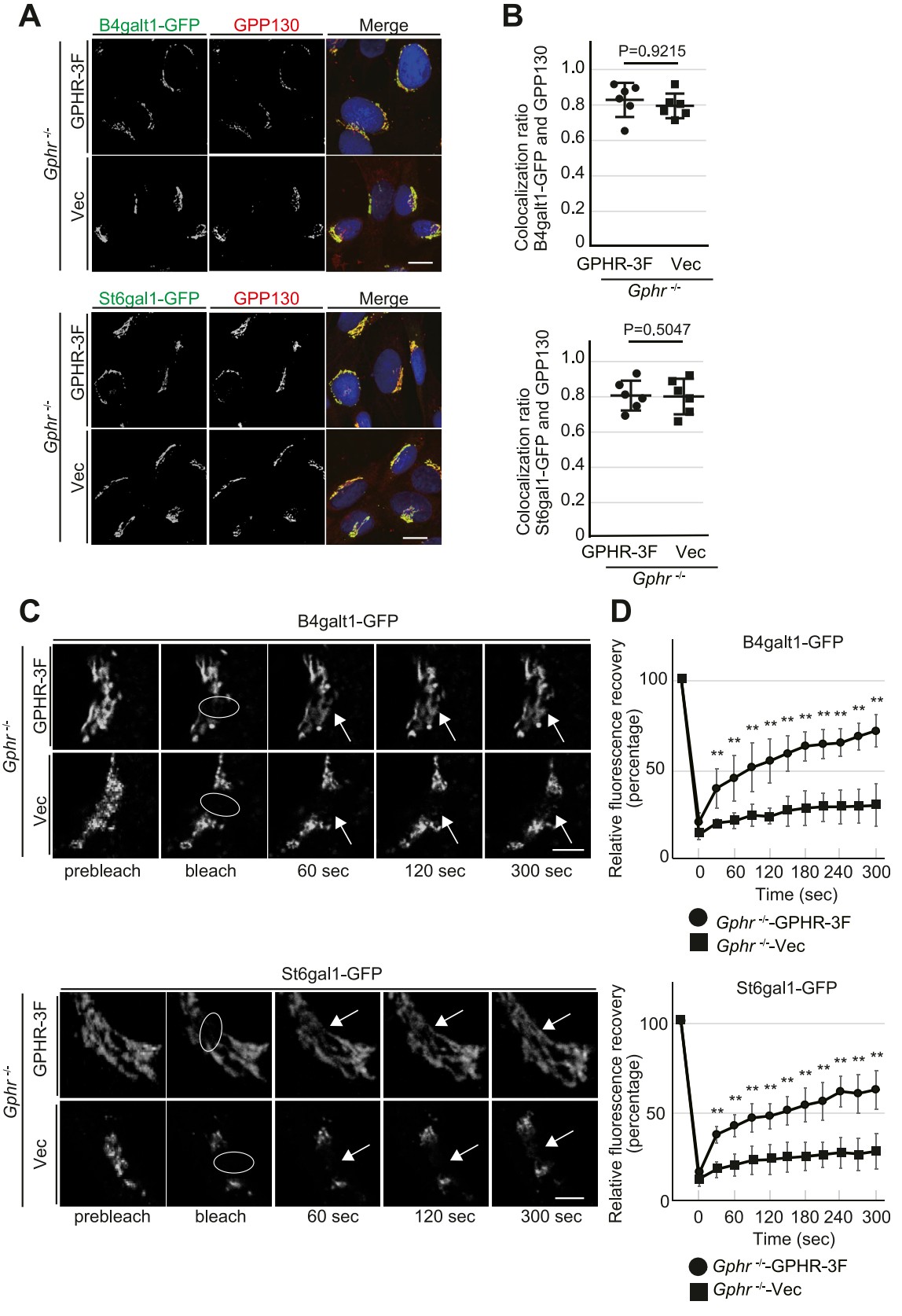

network, and B4galt1 antibodies. We also observed that the number and area of PLA signals strongly decreased in both MEFs (Fig S7E–H). These results suggested that *Gphr* deficiency leads to impaired proximity conditions between B4galt1 and St6gal1 in the Golgi apparatus. Moreover, combination with these results (Figs 2, 3, and 4) suggests that abnormal *N*-glycosylation may be a result of decreased proximity conditions caused by the loss of glycosyltransferase trafficking dynamics.

### Altered *N*-glycosylation of LMPs on lysosomal membrane in *Gphr*-deficient MEFs

One of the destinations for glycoproteins that undergo glycosylation in the Golgi apparatus is the lysosomal membrane (Klumperman, 2011). The lysosomal glycoprotein performs a protective function, and the inner side of the lysosomal membrane is covered with heavily glycosylated proteins known as LMPs (Saftig & Klumperman, 2009). Therefore, we investigated *N*-glycosylation of LMPs that are involved in lysosomal membrane integrity. To verify the global change in *N*-glycosylation because of *Gphr* deficiency, we conducted immunoblotting analysis on LAMP-1, LAMP-2, CD63, and LIMP-2, which are the most abundant LMPs (Saftig & Klumperman, 2009). Immunoblotting analysis revealed a markedly reduced LMP molecular weight in $Gphr^{-/-}$Vec MEFs compared with that in the $Gphr^{-/-}$GPHR-3F (Fig 5A). Treating cell lysates from both $Gphr^{-/-}$GPHR-3F and $Gphr^{-/-}$Vec MEFs with PNGase F resulted in LMPs with the same relative molecular weights, indicating that *N*-glycosylation was affected in $Gphr^{-/-}$Vec MEFs (Fig 5B). We also analyzed the glycosylation status of the LMPs using endoglycosidase H (Endo H), an enzyme that dissociates *N*-linked mannose-rich glycans; these glycans become Endo H resistant after modification by glycosyltransferases within the Golgi apparatus. Upon treatment with Endo H, the band corresponding to LMPs shifted downwards (Fig 5C). This effect was heightened in *Gphr*-deficient cells, suggesting incomplete *N*-glycosylation in the Golgi apparatus. After immunoblot analysis, we confirmed the altered *N*-glycosylation of LMPs by lectin staining for GS-II. GS-II lectin recognizes the terminal non-reducing *N*-acetyl-*D*-glucosaminyl residues of glycoproteins, which are only found in glycosylation intermediates of the Golgi apparatus. We performed combined GS-II staining and immunostaining for GM130, a Golgi apparatus marker. GS-II signals colocalized with GM130 in $Gphr^{-/-}$Vec and $Gphr^{-/-}$GPHR-3F MEFs; notably, GS-II signal was present in addition to GM130-positive structures in $Gphr^{-/-}$Vec MEFs (Fig 5D). Quantification confirmed that the co-localization rate between GS-II and GM130 was significantly reduced in $Gphr^{-/-}$Vec MEFs compared with that in the $Gphr^{-/-}$GPHR-3F MEFs (Fig 5E). Furthermore, the GS-II signals in $Gphr^{-/-}$Vec MEFs co-localized with the LAMP-1-positive structures,

indicating incomplete *N*-glycosylation of lysosomal membrane proteins (Fig 5F). Quantification confirmed that the co-localization rate between GS-II and LAMP-1 was significantly increased in $Gphr^{-/-}$Vec MEFs compared with that in $Gphr^{-/-}$GPHR-3F MEFs (Fig 5G). These results indicate that LMPs with incomplete glycosylation were localized to the lysosomal membrane in $Gphr^{-/-}$Vec MEFs.

### Glycosylation is important for the maintenance of lysosomal membrane stability

Lysosomes are acidic organelles with a luminal pH of ~4.5–5.0, which is essential for the optimal activity of lysosomal hydrolases. These hydrolases are responsible for degrading biological macromolecules and are delivered via the endocytic, phagocytic, and autophagic pathways (Saftig & Klumperman, 2009). We investigated whether the *N*-glycosylation of LMPs influences the pH and degradation mechanism of lysosomes. The pH-sensitive fluorescent probe fluorescein isothiocyanate (FITC)-dextran, together with the insensitive fluorescent probe tetramethylrhodamine-5-isothiocyanate (TRITC)-dextran, were used to confirm lysosomal luminal acidic conditions (Fig S8A). Treatment with $NH_4Cl$, a neutralizing agent for lysosomal compartments, resulted in increased fluorescence intensity in $Gphr^{-/-}$GPHR-3F and $Gphr^{-/-}$Vec MEFs (Fig S8A). Quantification analysis confirmed that the mean fluorescence intensity was not significantly different between $Gphr^{-/-}$GPHR-3F and $Gphr^{-/-}$Vec MEFs (Fig S8B). We then analyzed lysosomal protein degradation using an autophagic flux assay (Mizushima & Murphy, 2020). Immunoblotting analysis for autophagosome protein LC3-II and autophagic substrate p62/sqstm1 revealed no significant difference in protein levels between $Gphr^{-/-}$GPHR-3F and $Gphr^{-/-}$Vec MEFs (Fig S8C and D). Treatment with BafilomycinA1 (BafA1), an inhibitor of lysosomal acidification, significantly increased the protein levels of LC3-II and p62 in both MEF groups (Fig S8C and D). Taken together, these results suggest that proper *N*-glycosylation of LMPs does not affect lysosomal pH and degradation in $Gphr^{-/-}$Vec MEFs.

Next, we investigated whether proper *N*-glycosylation of LMPs contributes to lysosomal membrane stability. Membrane stability was evaluated based on the ability of lysosomes to retain LysoTracker, a weak base that accumulates in acidic lysosomes and is fluorescent at low pH (Chazotte, 2011). As expected, incubation of MEFs with terfenadine, a cationic amphiphilic drug that permeabilizes the lysosomal membrane (Aits et al, 2015), reduced LysoTracker staining of lysosomes in a time-dependent manner (Fig 6A). Quantification revealed a significant reduction in the number of LysoTracker dots in $Gphr^{-/-}$Vec MEFs 30 min after terfenadine treatment compared with that of $Gphr^{-/-}$GPHR-3F MEFs (Fig 6B). Transcription factor EB (TFEB), a master transcriptional regulator of

---

**Figure 3. Gphr deficiency influences glycosyltransferase trafficking dynamics.**
**(A)** Confocal microscopy image. MEFs stably expressing GFP fusion glycosyltransferases B4galt1 or St6gal1 were fixed and stained with the Golgi marker GPP130. Cell nuclei were stained with DAPI. Scale bar: 10 *μ*m. **(A, B)** Bar graph indicating the quantitative GFP/GPP130 co-localization ratio shown in (A). Data are means ± SD of individual values (*n* = 6 images, <40 cells examined). *P*-values were corrected using Welch's *t* test. Actual *P*-values for each condition are indicated. **(C)** Time-lapse microscopy analysis. MEFs stably expressing either B4galt1-GFP or St6gal1-GFP were photobleached. The recovery of fluorescence was observed using live-cell imaging. Scale bar: 2 *μ*m. **(D)** The ratio of fluorescence of the bleached area to that of an adjacent unbleached area was measured for each time point, normalized to the initial values, and plotted. Data are presented as mean ± SD of non-photobleached (*n* = 10) and photobleached (*n* = 10) samples. Significant differences (**$P$ < 0.01) between the MEFs are shown. *P*-values were corrected using Welch's *t* test. See also Video 1, Video 2, Video 3, and Video 4.

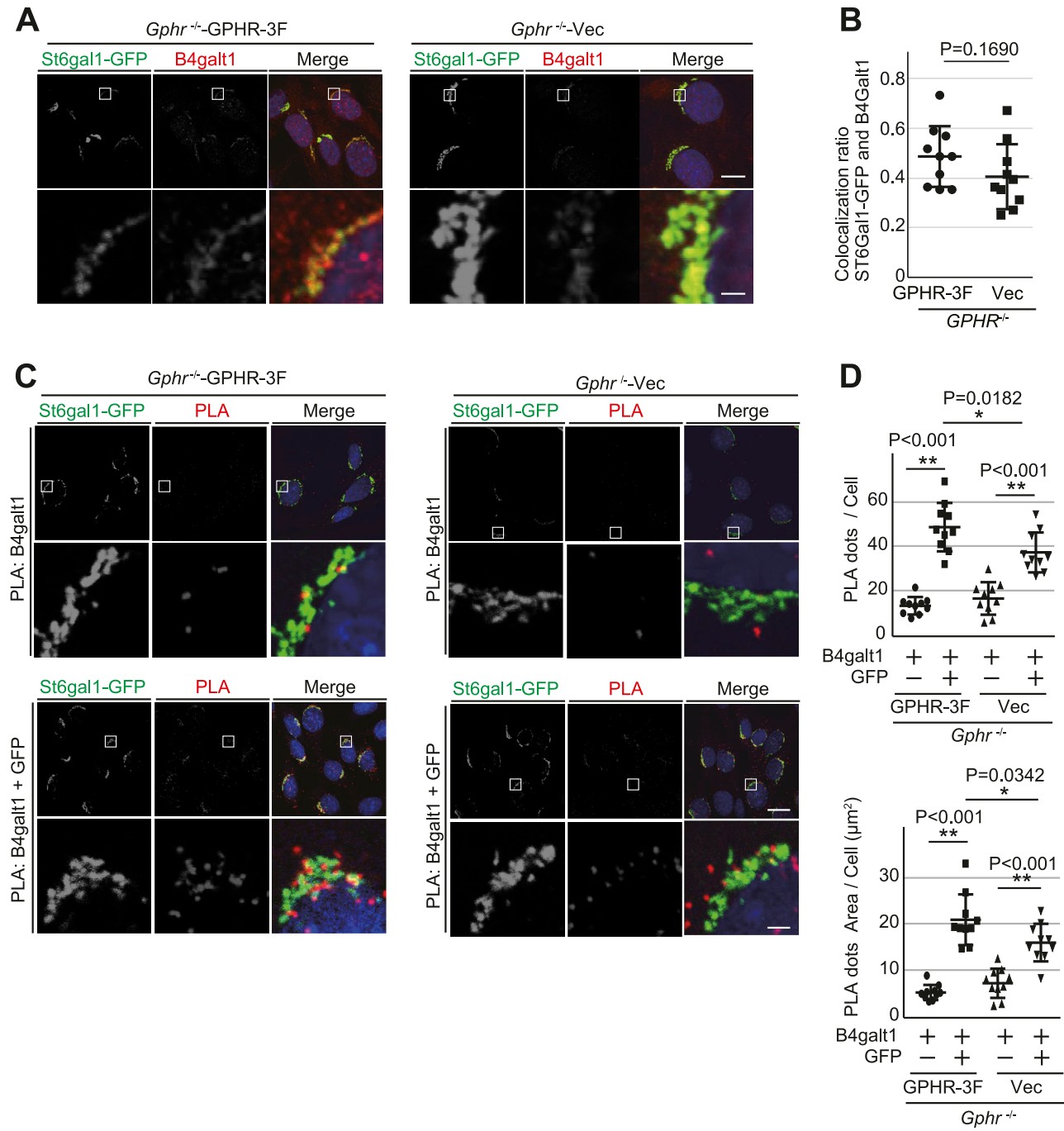

**Figure 4. Proximity ligation assay (PLA) of glycosyltransferases.**
**(A)** Confocal microscopy images showing St6gal1-GFP and endogenous B4galt1 expression in MEFs. Boxed regions are enlarged and shown in the lower panels. Cell nuclei were stained with DAPI. Scale bar: 10 and 1 $\mu$m (enlarged image). **(A, B)** Bar graph indicates the quantitative GFP/B4galt1 co-localization ratio shown in (A). Data are means ± SD of individual values ($n$ = 10 images, <90 cells examined). $P$-values were corrected using Welch's $t$ test. Actual $P$-values for each condition are indicated. **(C)** Representative images of PLA. Evidence of the proximity between B4galt1 and St6gal1-GFP is indicated by dots. B4galt1 was used as a negative control. Boxed regions are enlarged and shown in the lower panels. Cell nuclei were stained with DAPI. Scale bar: 10 and 1 $\mu$m (enlarged image). **(D)** Quantification of the PLA signal. Bar graphs indicate the quantitative number and area of the PLA dots for each condition. Data are means ± SD of individual values ($n$ = 10 images, <80 cells examined). Significant differences (*$P$ < 0.05, **$P$ < 0.01) and actual $P$-values for each condition are indicated. Outlier data points were defined as any value >1.5 times the interquartile range from either the lower or upper quartile, and $P$-values were corrected for one-way ANOVA with Tukey's *post hoc* test.

lysosomes, was reportedly translocated to the nucleus after lysosomal membrane damage (Sardiello et al, 2009; Chauhan et al, 2016; Jia et al, 2018). Therefore, we assessed the nuclear

translocation of TFEB after terfenadine treatment using immunostaining. Under untreated conditions, TFEB was diffusely distributed throughout the cytosol, and a few particles were distributed

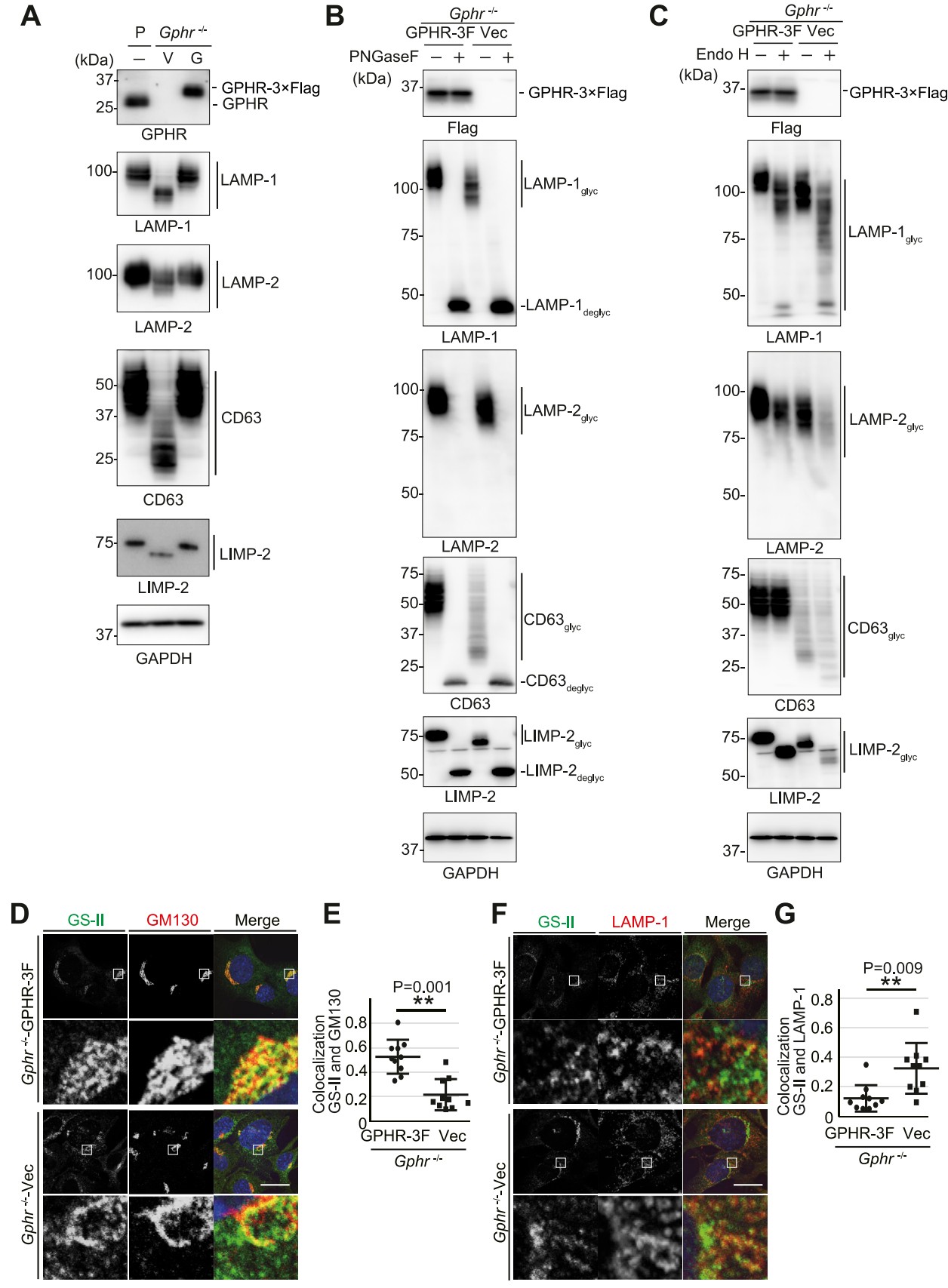

throughout the nucleus. However, TFEB was translocated to the nucleus after terfenadine treatment in both MEF groups (Fig 6C). Quantification confirmed that *Gphr*⁻/⁻Vec MEFs had significantly increased nuclear translocation of TFEB 60 min after terfenadine treatment compared with that in *Gphr*⁻/⁻GPHR-3F MEFs (Fig 6D). Next, we investigated whether terfenadine treatment caused lysosomal membrane damage to induce Galectin-3 recruitment. Galectin-3 is a cytoplasmic β-galactose-binding lectin that recognizes luminal glycans of glycoproteins when the lysosomal membrane is damaged (Paz et al, 2010; Maejima et al, 2013). We found that Galectin-3, which is diffused throughout the cytosol, was translocated to the LAMP-1-positive lysosome after terfenadine treatment (Fig 6E). Importantly, the number of Galectin-3 dots increased 120 min after terfenadine treatment in *Gphr*⁻/⁻Vec MEFs compared with that in *Gphr*⁻/⁻GPHR-3F MEFs (Fig 6F). These results suggest that *Gphr* deficiency affects lysosomal membrane stability and accelerates lysosomal membrane damage induced by terfenadine treatment. We also confirmed similar properties of lysosomal membrane in *Gphr*-deficient N2A cells (Fig S9A–E). Moreover, we verified the importance of *N*-glycosylation of LMPs in lysosomal membrane stability using kifunesine and swainsonine, inhibitors of alpha-mannosidase. Inhibition of glycosylation in LMPs was achieved by treatment with kifunesine or swainsonine (Fig S10A). In comparison with the DMSO control, the quantitative assessment of LysoTracker retention, TFEB nuclear translocation, and Galectin-3 dot assay results, induced by terfenadine, demonstrated significant differences in the inhibitor-treated MEFs (Fig S10B–G). Based on these results, we conclude that altered *N*-glycosylation of LMPs affects lysosomal membrane stability. In summary, our results support the hypothesis that luminal pH regulation of the Golgi apparatus by GPHR contributes to lysosomal membrane stability via the proper *N*-glycosylation of LMPs.

# Discussion

In this study, we demonstrated that GPHR regulates proper *N*-glycosylation by maintaining glycosyltransferase dynamics and proximity conditions in the Golgi apparatus. Glycomic analysis using TOF mass spectrometry revealed that loss of GPHR affects *N*-glycosylation in several steps. Importantly, FRAP of glycosyltransferase fused to GFP indicated that the luminal pH of the Golgi apparatus influenced the regulation of glycosyltransferase trafficking dynamics. PLA revealed that abnormal pH of the Golgi apparatus reduced co-localization of glycosyltransferases. Moreover, *Gphr* deficiency led to formation of incomplete *N*-type glycoproteins (LMPs), suggesting its importance in maintaining lysosomal membrane stability. Thus, pH regulation of the Golgi

apparatus contributes to the maintenance of the stability of lysosomal membrane proteins through precise *N*-glycosylation. Furthermore, we previously demonstrated that mutations in *Gphr* impaired protein glycosylation in CHO cells (Maeda et al, 2008). Thus, GPHR-mediated regulation of luminal pH of the Golgi apparatus could contribute to the maintenance of cellular *N*-glycosylation in different cell types.

The Golgi apparatus constantly produces a diverse array of glycan structures such as *N*-glycans, *O*-glycans, glycosaminoglycans (GAGs), glycosylphosphatidylinositol (GPI), and other glycans. Different glycosyltransferases are specialized for specific glycan classes, whereas certain enzymes are involved in synthesizing common structures shared among these classes. These enzymes exhibit competitive or cooperative interactions within the Golgi apparatus, influencing the development of cell-specific glycan patterns (Reily et al, 2019). The TOF mass analysis in the current investigation revealed that multiple steps of N-glycosylation were impaired in *Gphr*-deficient MEFs. Nevertheless, this approach was unable to detect *O*-glycans and larger structures, including glycans containing polylactosamine and GAGs. We could not rule out the potential impact of large glycan structures in this study; therefore, further analysis is needed to characterize the properties attributed to the new *Gphr* defect.

Sialylated oligosaccharide structures of mucin were altered by pH neutralization in the Golgi apparatus of ammonium chloride-treated LS 174T cells (Axelsson et al, 2001). Similarly, chloroquine-induced elevation of luminal pH of the Golgi apparatus inhibited *O*-glycosylation of mucin and terminal a-2,3-sialylation of *N*-glycosylation in MCF-7 and COS-7 cells (Rivinoja et al, 2006). In agreement with these studies, our glycomic and biochemical analyses indicated that the absence of GPHR in cells led to a significant decrease in the sialylation of *N*-glycans. These observations indicate that the sialylation of *N*-glycans is highly sensitive to changes in the luminal pH of the Golgi apparatus. This may be because sialyltransferases, such as St6gal, are located and function in the TGN, which has a lower pH (pH 6.0) within the Golgi apparatus (Casey et al, 2010; Reily et al, 2019). Reduced sialylation of *N*-glycans has been reported in molecules with deletions or loss-of-function mutations that regulate the luminal pH of the Golgi apparatus (Guillard et al, 2009; Galenkamp et al, 2020). These results indicate that luminal pH regulation of the Golgi apparatus is essential for sialylation of *N*-glycans.

Glycosylation is considered to be a pH-sensitive process in the Golgi apparatus. Deletion or loss-of-functional mutation in *ATP6V0A2*, *GPHR*, *SLC4A2*, and *NHE7*, which are responsible for luminal pH regulation of the Golgi apparatus, results in abnormal glycosylation (Maeda et al, 2008; Hucthagowder et al, 2009; Udono et al, 2015; Galenkamp et al, 2020; Khosrowabadi et al, 2021). In addition, it has been demonstrated that elevated luminal pH of the Golgi apparatus also causes abnormal glycosylation (Axelsson et al,

---

**Figure 5. Altered N-glycosylation of lysosomal glycoproteins in Gphr-deficient MEFs.**
**(A)** Cell lysates prepared from parental (P), *Gphr*⁻/⁻Vec (V) and *Gphr*⁻/⁻GPHR-3F MEFs (G) were subjected to immunoblotting for indicated antibody against lysosomal membrane proteins. **(B, C)** Alterations in lysosomal glycoprotein molecular weight in *Gphr*-deficient MEFs. MEF lysates were deglycosylated using PNGase F (B) or Endo H (C), before immunoblotting with the indicated antibodies. Anti-LAMP-2 immunoreactivity was lost after deglycosylation using PNGase F. **(D, F)** Combined fluorescent lectin immunostaining. **(D, F)** MEFs were labeled with GS-II lectin and counterstained with GM130 (D) or LAMP-1 (F). Cell nuclei were stained with DAPI. Scale bar: 10 µm. **(E, G)** Bar graphs indicate the quantitative co-localization ratio of GS-II/GM130 (E) and GS-II/LAMP-1 (G). Data are means ± SD of individual values (*n* = 10 images, <70 cells examined). Significant differences (\*\**P* < 0.01) and actual *P*-values for each condition are indicated. *P*-values were corrected using Welch's *t* test. Source data are available for this figure.

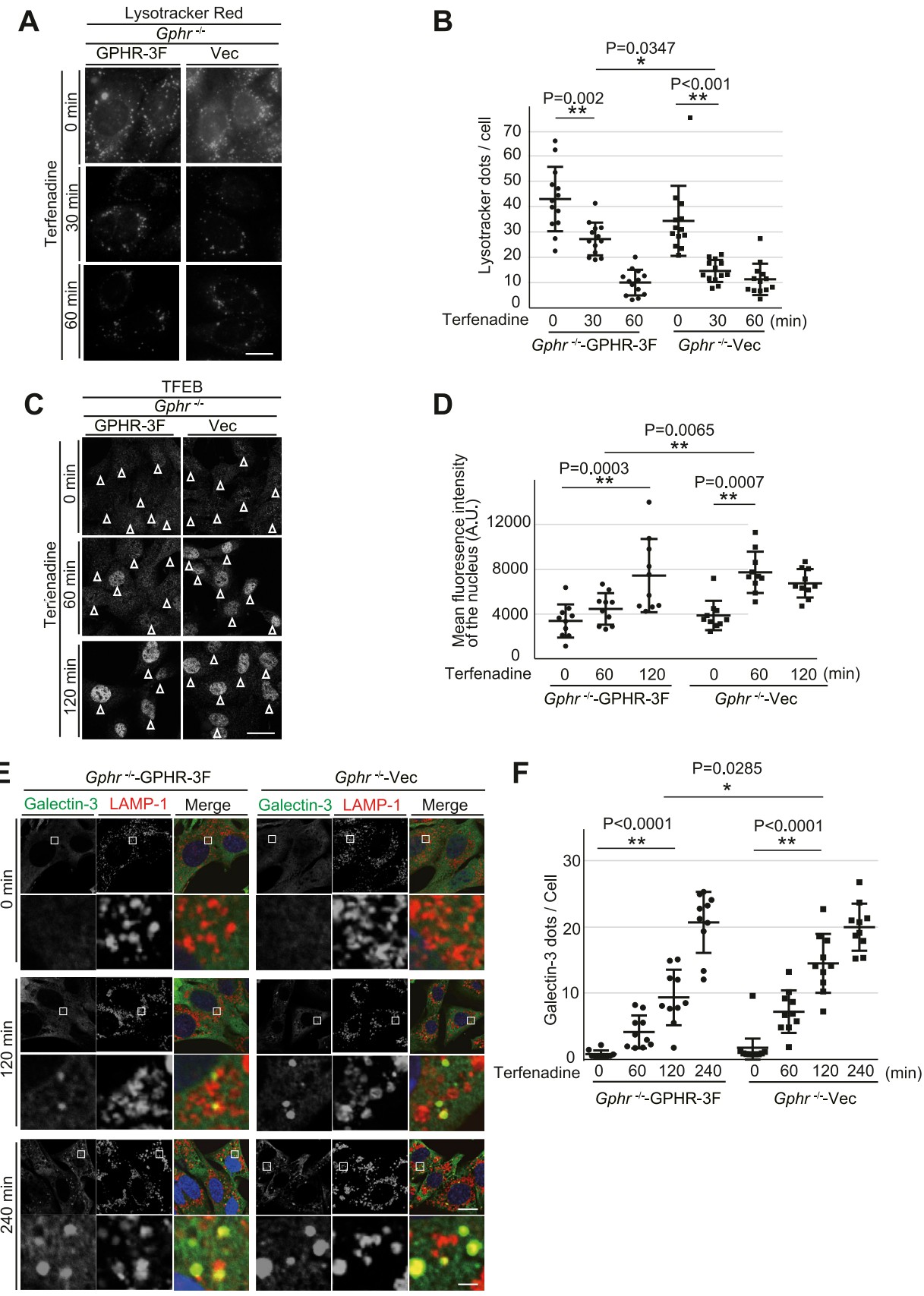

2001; Rivinoja et al, 2009). Although the precise mechanism by which dysregulated luminal pH of the Golgi apparatus impairs glycosylation is unclear, two possibilities have been proposed. One possibility is that the elevated pH is no longer within the optimal pH range for enzymatic activity; therefore, glycosylation is impaired. Another possibility is that elevated pH causes mislocalization of glycosyltransferases, resulting in the loss of its ability to catalyze reactions in the correct order. Impairment of acidification by compounds that have different mechanisms of action at acidic pH, such as BafA1, and ammonium chloride, has been shown to result in abnormal glycosylation of mucins because of incorrect localization of glycosyltransferases, such as N-acetylgalactosaminyltransferase 2 (GALNT2), N-acetylglucosaminyltransferase 1 (MGAT1), and B4GALT1 (Axelsson et al, 2001). However, we were unable to confirm the mislocalization of glycosyltransferases B4galt1 and St6gal1 in the Gphr-deficient cells (Fig 3). Interestingly, low-dose chloroquine treatment induced abnormal N-glycosylation because of mislocalization of N-acetyllactosaminide α(2,3)-sialyltransferase III (ST3GAL3) but not of ST6GAL1 (Rivinoja et al, 2009). Consistent with these observations, our results suggest that the Gphr deficiency does not cause an increase in pH sufficient to alter the localization of glycosyltransferases. In fact, the pH of Gphr-deficient cells tends to be lower than that of NH$_4$Cl-treated MEFs (Fig 1C). These results indicated that there was a difference in the effects observed in Gphr deficiency and those in cells treated with chemical compounds. Results of our FRAP analysis suggest that inhibition of the trafficking dynamics of glycosyltransferases within the Golgi apparatus is caused by elevated luminal pH because of Gphr deficiency (Fig 3). Moreover, glycosyltransferases have been known to undergo continuous retrograde transport from the late cisternae (trans) to early cisternae (cis) via COPI vesicles to maintain steady-state localization in the Golgi apparatus (Schmitz et al, 2008; Tu et al, 2008; Pellett et al, 2013; Liu et al, 2018). Vesicle trafficking also requires tight regulation of the luminal pH of the Golgi apparatus as it influences vesicle sorting and packaging (Casey et al, 2010). Supporting the role of trafficking dynamics in the Golgi apparatus, a recent study reported that Golgin45 protein levels regulate protein glycosylation through the Rab2-GTP-dependent trafficking dynamics of B4galt1 (Yue et al, 2021). Our PLA results (Fig 3) support the previous observation that increasing the luminal pH of the Golgi apparatus reduces the proximity condition of glycosyltransferases, although glycosyltransferase is retained in the Golgi apparatus (Axelsson et al, 2001; Hassinen & Kellokumpu, 2014). Taken together, the results of the previous and present studies suggest that regulation of luminal pH plays a key role in the interaction of glycosyltransferase through proper trafficking dynamics.

The morphological integrity of the Golgi apparatus correlates with its function including glycosylation (Petrosyan, 2019). In mammalian cells, glycosyltransferases are thought to be ubiquitously distributed in the Golgi apparatus, which are further subdivided into smaller zones. The localization of enzymes is an extremely important factor in the synthesis of glycans because the glycosyltransferase reaction cannot occur without the encounter between an enzyme and a substrate (Welch & Munro, 2019). Our EM analysis indicates that Gphr deficiency results in abnormal morphology of the Golgi apparatus (Figs 1G, S1D, and S2D). The morphological abnormalities impair the proper transport of glycosyltransferases, which in turn prevents the association of glycosyltransferases (Figs 3 and 4). Consequently, the decreased association of these glycosyltransferases reduces the opportunities for substrate delivery, which in turn leads to abnormal glycosylation. In addition, mutations in ATP6V0A2 and TMEM87A genes, which are involved in luminal pH regulation, have been reported to cause abnormal morphology and glycosylation because of pH abnormalities of the Golgi apparatus (Guillard et al, 2009; Kang et al, 2024). Further studies are needed to investigate how abnormal morphology of the Golgi apparatus is caused by abnormal pH regulation.

The lysosomal membrane is a physical barrier separating the acidic environment of the lumen from the cytoplasm that protects cellular components from lysosomal hydrolases (Saftig & Klumperman, 2009; Schwake et al, 2013). However, the mechanisms that protect the lysosomal membrane remain largely unknown. Glycosylation of the luminal portion of lysosomal membrane proteins is thought to be protective, forming a continuous glycan layer (glycocalyx) on the lysosomal luminal surface and thereby preventing lysosomal hydrolase-mediated membrane damage (Lewis et al, 1985; Wang et al, 1991). By inducing Gphr deficiency, we were able to generate abnormal N-glycosylation of LMPs in the lysosomal membrane (Fig 4). However, it did not affect lysosomal function, such as the maintenance of a highly acidic environment and proteolysis under physiological conditions. Nevertheless, lysosomal membrane permeabilization occurred in Gphr-deficient MEFs, as evidenced by the reduced number of LysoTracker dots after terfenadine treatment. In addition, there was an increase in the nuclear translocation of TFEB, which represents lysosomal damage response; TFEB is a master regulator of lysosomal biogenesis at the transcriptional level (Sardiello et al, 2009). The Galectin-3 assay is another useful method for monitoring lysosomal damage response (Paz et al, 2010). We observed an increase in the number of Galectin-3 dots, which is associated with lysosomal membrane damage. Therefore, our findings indicate that

**Figure 6. The role of N-glycosylation of lysosomal membrane proteins in lysosomal membrane stability.**
**(A)** Quantitative assessment of lysosomal acidification. MEFs were incubated with LysoTracker Red and terfenadine and fixed at different time points as indicated. Representative images are presented. Scale bar: 20 µm. **(B)** Quantification of the number of LysoTracker dots. **(A)** Bar graphs indicate the mean number of LysoTracker dots per cell shown in (A). Data are means ± SD of individual values (n = 10 images, <150 cells examined). **(C)** Intracellular distribution of TFEB in MEFs. MEFs were incubated with terfenadine and fixed at the indicated time points. MFEs were stained with an anti-TFEB antibody. Scale bar: 10 µm. **(D)** Intensity of TFEB in the nuclei. **(C)** Bar graphs indicate the quantified fluorescence intensity of TFEB in the nuclei shown in **(C)**. Data are means ± SD of individual values (n = 10 images, <200 cells examined). **(E)** Confocal microscopy images. MEFs were incubated with terfenadine, fixed at the indicated time points, and stained with anti-Galectin-3 and anti-LAMP-1 antibodies. Boxed regions are shown in the under panels. Cell nuclei were stained with DAPI. Scale bar: 10 and 1 µm (enlarged image). **(F)** Quantification of the number of Galectin-3 dots per MEF under each condition. Bar graphs indicate the number of Galectin-3 dots. Data are means ± SD of individual values (n = 10 images, <70 cells examined). For (B, D, F), significant differences (*P < 0.05, **P < 0.01) and actual P-values for each condition are indicated. P-values were corrected for one-way ANOVA with Tukey's post hoc test.

proper glycosylation of LMPs contributes to the formation of a glycocalyx, which plays a role in preserving lysosomal membrane stability.

Maintaining luminal acidic pH is crucial for the integrity of the Golgi apparatus, particularly for protein glycosylation. Alterations in glycosylation have been linked to various diseases including congenital disorders of glycosylation, neurological disorders, and cancer (Reily et al, 2019). *ATP6V0A2* encodes the α2 subunit of V-ATPase, which is localized in the Golgi apparatus and is associated with cutis laxa type II. Loss-of-function mutations in *ATP6V0A2* lead to impaired protein expression and defects in the sialylation of both *N*- and *O*-glycans (Kornak et al, 2008). Loss of Ube3a/E6AP, the causative gene for Angelman syndrome, results in abnormal acidification of the Golgi apparatus and protein sialylation (Condon et al, 2013). Notably, glycosylation of the lysosomal glycocalyx was altered in Niemann–Pick disease type C model cells (Li et al, 2015; Kosicek et al, 2018). Further glycomic analysis of the lysosomal membrane proteins in Golgi-related diseases, and congenital disorders of glycosylations, provides a better understanding of the function of the lysosomal glycocalyx and reveals its potential pathological function in lysosomal dysfunction.

In conclusion, our study provides valuable insights into the interconnected roles of luminal pH regulation in the Golgi apparatus, protein glycosylation, and lysosomal membrane stability. Regulation of luminal pH in the Golgi apparatus is of critical importance for the proper protein glycosylation of proteins, which occurs through the interaction of glycosyltransferases. Glycosylation of LMPs contributes to lysosomal membrane stability and potentially protects the membrane from lysosomal damage. Further studies are necessary to elucidate the detailed mechanisms underlying the association between protein glycosylation and lysosomal membrane stability.

# Materials and Methods

### Cell culture

MEFs were obtained from 13.5-d-old embryos. Immortalized *Gphr*$^{flox/flox}$ MEFs were established by infecting a recombinant retrovirus carrying a temperature-sensitive simian virus 40 large T antigen (DeCaprio et al, 1988). Parental and *Gphr*-deficient MEFs were maintained in DMEM containing 10% FBS (Thermo Fisher Scientific), 5 U/ml penicillin, and 50 μg/ml streptomycin (Fujifilm Wako Pure Chemical Corporation). To establish GPHR-deficient MEFs, *Gphr*$^{flox/flox}$ MEFs were transfected with Cre recombinase using the adenovirus expression system. 2 d after transfection, MEFs were diluted by the culture medium to select clonal *Gphr*-deficient MEFs. GPHR deficiency was confirmed by immunoprecipitation and immunoblotting with an anti-GPHR antibody. Mice were housed under specific pathogen-free conditions with 12/12 h light/dark cycles at Juntendo University. The procedures involving animal care, surgery, and sample preparation were approved by the Animal Experimental Committee of Juntendo University and were conducted in accordance with the guidelines for the Care and Use of Laboratory Animals. To establish the *Gphr*-deficient N2A (known as Neuro 2A) cell line, the *Gphr* guide RNA was designed using the

CRISPR Design Tool (Horizon Discovery Ltd) and subcloned into pX330-U6-Chimeric_BB-CBh-hSpCas9 (#42230; Addgene). N2A cells were transfected with the pX330 vector containing the target gRNA using Lipofectamine 2000 (Thermo Fisher Scientific). 2 d after transfection, N2A cells were diluted to select clonal *Gphr*-deficient N2A cells. *Gphr* deficiency was confirmed by immunoprecipitation and immunoblotting with an anti-GPHR antibody. For the LysoTracker assay, cells were cultured with 100 nM LysoTrackerTM Red DND-99 (Thermo Fisher Scientific) for 30 min, chased in normal medium for 30 min, and treated with or without 10 μM terfenadine (Merck).

### Antibodies and reagents

For immunoblotting, the following antibodies were used: mouse monoclonal anti-GAPDH (MAB374; Merck), rabbit monoclonal anti-CD63 (clone EPR21151; Abcam), CD71 (clone H68.4; Cell Signaling Technology), rabbit monoclonal anti-TGN38 (clone E2T4P; Cell Signaling Technology), rabbit anti-GPP130 (923801; BioLegend), mouse monoclonal anti-FLAG (M2; Merck), rabbit monoclonal anti-LAMP-1 (clone C54H11; Cell Signaling Technology), rat monoclonal anti-LAMP-2 (clone GL2A7; StressMarq Biosciences), rabbit monoclonal anti-LIMP-2 (clone E2Z5F; Cell Signaling Technology), anti-mouse monoclonal anti-MAN2A1 (clone F-10; Santa Cruz Biotechnology), rabbit polyclonal anti-B4galt1 (11220-RP02; Sino Biological), and rabbit polyclonal anti-St6gal1 (20485-1 AP; Proteintech). The preparation of monoclonal mouse anti-GPHR was previously described (Maeda et al, 2008).

For immunofluorescence assays, the following antibodies were used: mouse monoclonal anti-GM130 (610822; BD), rat monoclonal anti-LAMP-1 (clone 1D4B; Santa Cruz Biotechnology), rabbit anti-GPP130 (923801; BioLegend), goat anti-B4galt1 antibody (AF3609; R&D Systems), and rabbit monoclonal anti-TGN38 (clone E2T4P; Cell Signaling Technology). For lectin blotting and lectin staining, the following lectins were used: Biotinylated SNA lectin (B-1305-2; Vector Laboratories) and Alexa Fluor 488 conjugate GS-II (L21415; Thermo Fisher Scientific).

### Plasmids

To construct pMXs-Puro B4galt1-GFP and pMxs-Puro St6gal1, cDNA fragments of B4galt1 and St6gal1 were amplified by PCR and inserted into pEGFP-N1 (Takara Bio). B4galt1-GFP and St6gal1-GFP were amplified using PCR and inserted into pMXs-puro (Cell Biolabs Inc., CA, USA). PCR amplification was performed using KOD FX neo (TOYOBO).

### Virus vector system

To establish stably expressing GPHR-3xFlag cells, we used a retrovirus vector system (Cell Biolabs Inc.). Platinum-E cells were transiently transfected with pMXs-BSD-GPHR-3xFlag vectors using Lipofectamine 2000 (Thermo Fisher Scientific), and the medium containing the retrovirus was collected. *Gphr*-deficient MEFs were incubated in a virus-containing medium supplemented with 1 μg/ml polybrene. For the expression of Cre recombinase, we used an Ad-CMV-iCre adenovirus expression system (1045; Vector Biolabs).

The cells were seeded at a density of $1 \times 10^5$ cells/well in 6-well dishes. After 24 h, a culture medium containing adenovirus was added to the medium at a multiplicity of infection of 10. 2 d after infection, MEFs were diluted to select clonal *Gphr*-deficient MEFs.

## Electron microscopy

Cells were fixed with 2% PFA and 2% glutaraldehyde in 0.1 M phosphate buffer (pH 7.4) at 4°C overnight. Subsequently, they were post-fixed in 2% osmium tetroxide at 4°C for 2 h, block-stained with 0.5% uranyl acetate for 30 min, and embedded in Epon812 (Oken Shoji). Ultrathin sections (80 nm) were cut with an ultramicrotome UC6 (Leica Microsystems) and stained with uranyl acetate and lead citrate. The sections were then observed with an electron microscope (EM; JEM1400Plus; JEOL).

## Immunoblotting

MEFs were lysed in ice-cold lysis buffer (50 mM Tris–HCl [pH 7.5], 150 mM NaCl, 1% TX-100, 1 mM EDTA, and protease inhibitor cocktail [Merck]). Lysates were normalized using a bicinchoninic acid protein assay kit (Thermo Fisher Scientific). Protein samples were subjected to sodium dodecyl sulfate-polyacrylamide gel electrophoresis (SDS–PAGE) and transferred to polyvinylidene difluoride (PVDF) membranes (Merck). The PVDF membranes were blocked with 5% skim milk in TBST buffer (20 mM Tris–HCl [pH 7.5], 150 mM NaCl, 0.1% Tween 20) for 30 min at 24°C. After incubation overnight at 4°C with the primary antibodies (1:500 dilution), the PVDF membranes were incubated with horseradish peroxidase-conjugated goat anti-mouse IgG (H + L) or goat anti-rabbit IgG (H + L) (1: 10,000 dilution) (115-035-166 and 111-035-144; Jackson ImmunoResearch Laboratories, Inc.) for 30 min at 24°C. SuperSignal West Pico (Thermo Fisher Scientific) was used for the detection. Densitometry analysis was performed using Multi Gauge V3.2 (FUJIFILM Corporation).

## Immunofluorescence and microscopy

Cells were fixed with 4% PFA for 15 min at 25°C. They were then washed with PBS, permeabilized with 0.1% TX-100 or 50 ng/ml digitonin in PBS for 5 min, and blocked with TNB blocking buffer (PerkinElmer) for 30 min. The cells were then incubated with a mixture of primary antibodies (1:200 dilution) in TNB blocking buffer for 60 min at 25°C, followed by incubation with fluorescently labeled secondary antibodies (1:500 dilution; Alexa Fluor 488 [Thermo Fisher Scientific], Cy3, and Cy5 [Jackson ImmunoResearch]) for 30 min at 25°C. Fluorescence images were obtained using an FV1000 confocal laser-scanning microscope (Olympus). The Hybrid Cell Count Software (Keyence) was used to quantify the images.

## FRAP

For FRAP analysis, MEFs stably expressing B4galt1-GFP or St6gal1-GFP were cultured in a normal culture medium in an atmosphere of 5% $CO_2$ at 37°C. The MEFs were imaged using a 100× objective on an FV1000 confocal laser-scanning microscope. Part of the Golgi apparatus was bleached using a single laser pulse. Images were acquired every 15 s for 7 min. Fluorescence values in the bleached

and recovered areas were measured using Fiji software (Schindelin et al, 2012). Fluorescence recovery was normalized to pre-bleaching and post-bleaching values. The experiments were repeated twice, and graphs were plotted using the average values of 10 experiments.

## Glycomic analysis

Each sample was analyzed to quantify *N*-linked glycans using the S-Bio proprietary GlycanMap Xpress (Sumitomo Bakelite Co., Ltd.) methodology. Several control samples were analyzed to assess linearity, repeatability, sensitivity, and accuracy. For quantitation, internal standards were added to each sample. Samples were denatured and digested with trypsin, followed by heat inactivation. The *N*-glycans were then enzymatically released from the peptides by treatment with PNGase F (New England Biolabs), and the released glycans were subjected to solid-phase processing using BlotGlyco beads (Sumitomo Bakelite Co. Ltd.). After capture on the beads, the sialic acid residues were methyl-esterified to stabilize them in the mass spectrometer. Glycans were simultaneously released from the beads and labeled, and aliquots of the recovered materials were spotted on a MALDI target plate. The procedure, from initial aliquoting to spotting on the MALDI plate, was performed using fully automated, 96-well format, robotic technology.

MALDI-TOF mass spectrometric analysis was performed using an Ultraflex III mass spectrometer (Bruker Daltonics) in positive-ion reflectron mode using a proprietary matrix composition. Each sample from the bead-based processing step was spotted in quadruplicate, and spectra were obtained in an automated manner using the AutoXecute feature in the flexControl software (Bruker Daltonics). Mass spectra were analyzed using S-BIO's proprietary bioinformatics programs. The data analysis methodologies for the GlycanMap Xpress assay contain algorithms to correct for additional minor species with different masses resulting from the chemical derivatization of the parent glycan.

## PLA

PLA was performed using a PLA kit (Duolink In Situ; Merck) according to the manufacturer's instructions. Briefly, the fixed cells were blocked using the blocking solution provided in the kit for 30 min at 37°C. Primary antibodies (goat anti-B4Galt1 and mouse monoclonal anti-GFP [5G4, Cell Signaling Technology]) diluted (1:200) in the antibody diluent included in the kit were added to the samples and incubated overnight at 4°C. PLA probes (anti-goat PLUS and anti-mouse MINUS) were then added and incubated for 1 h at 37°C. Ligation was performed for 30 min at 37°C using the ligase provided in the kit. Amplification using the polymerase provided in the kit was performed for 100 min at 37°C. Nuclear staining was performed using a mounting medium with DAPI. Finally, the cells were visualized using an FV1000 confocal laser-scanning microscope. Hybrid Cell Count Software (Keyence) was used to quantify the images.

## Statistical analysis

All results are presented as bar graphs. The unpaired *t* test (Welch's *t* test) was used to compare the means of two independent groups.

One-way analysis of variance (ANOVA) followed by Tukey's multiple comparison test was used to compare means among three or more groups. Differences observed with values of $P < 0.05$ were considered statistically significant. All statistical analyses were performed using GraphPad Prism 7 (GraphPad Software Inc.).

## Data Availability

Data supporting the results of this study are available from the corresponding author upon reasonable request.

## Supplementary Information

## Acknowledgements

We thank Dr. Shun Kageyama (Institute for Advanced Biosciences, Keio University) and Hirosi Kameda (Department of Cell Biology and Neuroscience, Juntendo University Graduate School of Medicine) for their helpful comments and discussion. We would like to thank Editage for English language editing. This work was supported by a Grant-in-Aid for Scientific Research from the Japan Society for the Promotion of Science and the Private School Branding Project from the Ministry of Education, Culture, Sports, Science and Technology of Japan (19K07276 and 23K062324 to Y-s Sou; 23K06323 to M Koike), and subsidies for the ordinary expenses of private schools from The Promotion and Mutual Aid Corporation for Private Schools of Japan. This study was also supported by a grant from the Institute for Environmental & Gender-specific Medicine, Juntendo University.

### Author Contributions

Y-s Sou: conceptualization, data curation, formal analysis, funding acquisition, investigation, project administration, and writing—original draft, review, and editing.
J Yamaguchi: formal analysis, investigation, and writing—review and editing.
K Masuda: data curation, formal analysis, and validation.
Y Uchiyama: supervision and investigation.
Y Maeda: resources, supervision, and writing—review and editing.
M Koike: resources, supervision, funding acquisition, and writing—review and editing.

### Conflict of Interest Statement

The authors declare that they have no conflict of interest.

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
