## [Reviewer comments · Life Science Alliance]

Life Science Alliance

Golgi pH homeostasis stabilizes the lysosomal membrane through N-glycosylation of membrane proteins

Yu-Shin Sou, Junji Yamaguchi, Keisuke Masuda, Yasuo Uchiyama, Yusuke Maeda, and Masato Koike

DOI: <https://doi.org/10.26508/lsa.202402677>

Corresponding author(s): Yu-Shin Sou, Juntendo University School of Medicine

Review Timeline:

Submission Date:	2024-02-25
Editorial Decision:	2024-03-25
Revision Received:	2024-06-21
Editorial Decision:	2024-07-15
Revision Received:	2024-07-19
Accepted:	2024-07-19

Transaction Report:

March 25, 2024

Re: Life Science Alliance manuscript #LSA-2024-02677-T

Dr. Yu-Shin Sou
Juntendo University School of Medicine
Department of Cell Biology and Neuroscience
Hongo 2-1-1
Bunkyo-ku, Tokyo 113-8421
Japan

Dear Dr. Sou,

Thank you for submitting your manuscript entitled "Golgi pH homeostasis stabilizes lysosomal membrane through N-glycosylation of membrane proteins" to Life Science Alliance. The manuscript was assessed by expert reviewers, whose comments are appended to this letter. We invite you to submit a revised manuscript addressing the Reviewer comments.

Thank you for this interesting contribution to Life Science Alliance. We are looking forward to receiving your revised manuscript.

Sincerely,

B. MANUSCRIPT ORGANIZATION AND FORMATTING:

Reviewer #1 (Comments to the Authors (Required)):

In the present study, the authors generated and then used a MEF line deficient in Gpnr, encoding a Golgi proton counter ion transporter previously reported to lead to an increased luminal pH in the Golgi apparatus. In line with previous observations, using glycomics and gel-shift assays the study revealed that loss of GPHR in cultured MEFs leads to abnormal glycosylation patterns. Using confocal microscopy and live-cell imaging, the authors also assessed the localization and trafficking of Golgi enzyme reporter constructs and spotted differences in Golgi FRAP and proximal interactions of Golgi enzymes between Gpnr-proficient and -deficient MEFs. They then focus specifically on lysosomal membrane proteins that - not unexpectedly - are also affected by altered glycosylation in Gpnr-deficient cells. Following drug-induced lysosomal damage/stress, the authors find that Gpnr-deficient and -proficient cells display distinct stress response dynamics and come to the conclusion that proper Golgi N-glycosylation is important for lysosomal membrane integrity.

In general, the study is well executed and scientifically sound. It does, however, have a number of imperfections that need to be addressed before I consider this manuscript suitable for publication. Moreover, the scientific implications of the findings presented have a limited scope and would need to be further substantiated to meet the expectations for manuscript suitable for publication in Life Science Alliance.

Major points:

The authors present three distinct GPHR KO clones tested by western blot. In some experiments GPHR KO #11 is further characterized. Is this the clone that was used to generate the GPHR (+) and GPHR (-) cells (i.e. stably transfected with plasmid and empty plasmid, respectively) used in most panels? That should be made clear somewhere. In some cases the authors also use the term "control MEFs". Which cells are meant with that? Parental MEFs with endogenous GPHR or GPHR KO MEFs stably transfected with the empty plasmid?

Related to that, a limitation of the study is certainly that it is (seemingly) done in one single KO clone and using a Cre-dependent KO system. It needs to be excluded that this is a clonal effect and it would also be of interest if the Golgi phenotype the authors report also holds true in other systems. I feel that the study would benefit from an independently generated CRISPR/Cas9 KO line (possibly a human cell) to further substantiate at least some of the authors' findings.

The authors do in fact observe a very substantial glycosylation phenotype (in particular, when looking at the gel shift assays) in their KO model. They do see differences in regards to the localization/dynamics of glycosyltransferases that might explain this effect. A more straightforward explanation could however also be that Golgi enzymes are somewhat destabilized in a Golgi apparatus with a dysregulated luminal pH and a reduced extent of glycosylation could simply be a consequence of reduced glycosyltransferase levels in the Golgi apparatus. This needs to be tested but cannot be done on ectopically expressed reporter constructs. Instead, the authors should have a close look (e.g. by immunoblotting) at the protein levels of cell-endogenous Golgi enzymes (such as B4gal1 and st6gal1 but possibly also other) in WT vs. GPHR KO cells.

The authors observe a substantial glycosylation phenotype and given the profound importance of glycosylation as a modification of proteins and lipids of the eukaryotic pathway this can absolutely be expected to have far-reaching physiological implications for many cellular processes. In the second part of the study the authors then assess the GPHR Kos glycosylation impact specifically in the context of lysosomal membrane proteins. The connection of these two aspects of the story is somewhat "bumpy", though and could possibly be improved for the benefit of an interested reader.

The authors performed various experiments to assess lysosomal membrane integrity (lysotracker retention, TFEB activation, and Galectin-3 recruitment) following treatment of the cells with the cationic amphiphilic drug terfenadine. I think the wording of the description of the data should be changed as also GPHR-proficient cells (i.e. cells with an intact lysosomal glycocalyx) do display lysosomal damage and are absolutely not protected. The authors can, however, quantify a statistically significant difference in stress response parameters between GPHR-deficient and -proficient cells. But the findings are not adequately described and the authors do not come to a specific enough conclusion whether GPHR does benefit lysosomal membrane integrity or not. The wording should be made clearer here. Also, a dose-response response experiment could be better suited to

visualize whether GPHR- cells are more sensitized to terfenadine-induced lysosomal membrane damage.

The authors address the role of the lysosomal glycoalyx in the context of the GPHR KO model. But these findings obviously are not limited to this model and should similarly hold when Golgi glycosylation is impaired in other ways. To substantiate the generalized importance of complex Golgi N-glycosylation for lysosomal membrane integrity, similar experiments need to be performed in cell culture models that display glycosylation defect due to other molecular defects. This could for instance be possibly done in cells deficient for key N-glycosylation enzymes (e.g. MGAT1 KO) or simply cells treated with N-glycosylation inhibitors such as kifunensine or swainsonine.

Minor points

The KO validation is provided in Suppl. Fig. 1 - but in Fig. 1A data are presented from KO cells rescued with an empty plasmid or a GPHR-3xFlag plasmid. Thus, it is not a comparison of WT and KO cells. That needs to be made clearer in the figure. In fact, a GPHR western blot (like the ones found in panel C and E) needs to be provided for panel 1A, to make it absolutely transparent to the reader right at the beginning which model is used.

Related to that, the rescue cells do actually express substantially more GPHR than the parental MEFs, but e.g. in terms of glycosylation they do not show any differences to parental cells (which is good). Would an overexpression be actually expected to lower luminal pH further?

Re: Complex formation of glycosyltransferases: The wording should be changed when it comes to describing their PLA findings. A PLA signal alone is not sufficient to claim two molecules actually form a stable complex. They are certainly in close proximity, though.

Citing rather old literature that used chemical/pharmacological compounds that alter organellar pH in cell culture models, the authors stress the advantage of KO models as they could more specifically alter pH in specific organelles such as the Golgi apparatus without having a rather broad impact. While this is a valid point, KO models do carry the risk that cells phenotypically adapt and thus could lead to artefactual observations. This risk should be discussed and, given that only one single clonal Gphr KO line was used in the study, it would be interesting to what extent the authors observations hold true in an acute depletion setting, e.g. shortly following genome editing in CRISPR/Cas9-targeted Gphr KO pools or following siRNA-mediated Gphr targeting.

Denomination of glycosyltransferases is inconsistent as sometimes the gene name is used (a useful way of naming of the glycosylation enzymes these days) (e.g. B4galt1) and sometimes the more conventional enzyme naming (e.g. GnT1 for Mgat1). This should be done consistently throughout the manuscript. In case of the glycosyltransferases the correct notation for mouse gene names is used. But at the same time the authors always refer to their cell model as GPHR KO - which is not the correct notation of a mouse gene name.

Missing data panels and incorrect data panel referencing in figure legends:

Based on the legend, panel D should contain immunostainings of the Golgi apparatus; that panel is missing. The legend lists electron micrographs for panel E, though. The missing data should be included and the main text should be changed to also refer to both panels. In light of this imperfection, the authors should carefully re-check all figure panels and the corresponding legends again to make sure that all data sets are accurately described.

There is a similar issue with Fig. 1, in the legend a description of panel B is named as panel C, panel D mentioned in the legend should be corrected to C and so on.

Electron microscopy: Not mentioned in the main text, the electron micrographs point to a substantially alter Golgi morphology. It would be good to describe this in a bit more detailed manner in the main text and also to label the images to help readers not well acquainted with ultrastructural images spot important differences

Preferably, individual data points should be used instead of box-plots.

Fig. 5A - lysotracker is misspelled

Reviewer #2 (Comments to the Authors (Required)):

This paper reports an examination of the effects of removing from mouse fibroblasts the Golgi membrane protein Golgi pH regulator (GPHR). The Golgi is known to be acidified, especially toward the trans side of the Golgi stack, and previous work has shown that GPHR plays a role in this process and so when it is deleted, Golgi pH is perturbed. Previous work from the authors has shown that in the absence of GPHR, the Golgi processing of protein glycans is perturbed with some modifications becoming

less prevalent. This new paper extends this work by focusing on the abundant glycoproteins that are resident in lysosomes and are thought to help protect the lysosomal membrane against the harsh environment of the lysosomal interior. They present convincing evidence that the glycosylation of the various lysosomal associated membrane proteins is substantially affected when GPHR is deleted, along with a concomitant perturbation of Golgi structure and diffusion of Golgi enzymes within the Golgi stack. They go on to show that the membranes of lysosomes are more susceptible to damage in the mutant cells when compared to wild-type. The effects are relatively small, but this is not unexpected given that GPHR loss causes only partial loss of glycan modifying activity, and the effects they do see are carefully quantified and demonstrated with multiple well-established assays.

This paper thus extends previous work on the importance of GPHR and hence Golgi pH for glycosyltransferase activity in the Golgi. It provides valuable evidence that defects in the Golgi can indirectly effect lysosome integrity which is of use for interpreting the effects of gene knockout experiments. There are a couple of relatively minor points that I noticed, but if these are addressed then I would be supportive of publication.

A) The authors should note somewhere in the text that the gene name for GPHR is Gpr89 (GPR89 in humans).

B) The authors assay glycans using mass-spectrometry (Fig, 1A and Table 1). However, this method can only detect a subset of glycans and will miss those that are very large such as poly-lactosamine and glycosaminoglycans. These may well be affected, and so the authors should note this limitation to the assay.

C) The immunofluorescence images that are single channel should be shown as gray-scale with only the merged images in colour.

Reviewer #3 (Comments to the Authors (Required)):

The manuscript by Sou et al investigates the consequences of altered pH homeostasis in the Golgi on glycan processing and the function of lysosomal glycoproteins. For this they delete the protein leak channel GPHR in mouse fibroblasts and perform cell biological and glycobiochemical experiments on the Golgi as well as characterisation of lysosomal proteins and lysosomal membrane integrity. The manuscript is well written and clearly laid out. The findings are consistent with previous literature on pH effects on Golgi glycosylation and the functional models of the glycocalyx in the lysosome lumen. The pH regulation from the point of view of GPHR is novel and this is an interesting addition to the glycosylation literature. I would recommend that the authors consider the following revisions before publication.

1. I feel that the evidence presented for pH alteration affecting glycosyltransferase dimerization is not compelling. The PLA assay, as the authors say, has an effective distance of ca. 40 nm. This is a reasonably long distance in the Golgi, could even span the distance between two adjacent cisternae. The PLA signal in WT cells is therefore not very strong evidence for direct interaction of enzymes; a negative control of a trans-Golgi enzyme that does not show a PLA signal would be nice to see. Moreover, given the observed disruption of Golgi architecture in the GPHR- cells, it is possible that the observed loss of PLA signal is due to enzymes in proximity of each other (but not in a complex) in the same or adjacent cisternae being moved further from each other via fragmentation of the organelle.

2. It would be good to include more information from the literature where altered pH has already been shown to alter glycosylation and enzyme localisation (e.g. for the Golgi ATPase mutations from patients). Given that the experiments described here are using a different pH regulator, this would I feel be complementary to the current study.

3. It is worth noting that there is a roughly similar amount of non-Golgi staining in figure 4D for the GPHR+ and GPHR- cells shown, suggesting that non-Golgi localisation of GSII staining is not a GPHR- only case. With this in mind, figure 4E showing a different confocal section, which does not seem to contain the Golgi and has a large amount of extra GSII staining not only in GPHR- but also in GPHR+ cells. In GPHR+ cells there is a pre-nuclear GSII staining cluster that does not look like the Golgi (it is rather punctate) - The authors may want to comment about what compartment this may be?

Minor points:

1. The abbreviation for lysosomal membrane proteins is inconsistent through the manuscript and the abstract.

2. In table 1 some of the drawn N-glycan structures are not possible - eg hybrid with 6 mannoses and a glycan with 4 mannoses

Please find below, detailed responses to the reviewer's comments.

Response to reviewer 1:

We express our appreciation to the reviewers for their insightful comments on our paper. The comments helped us to significantly improve the paper.

Reviewer #1 (Comments to the Authors (Required)):

In the present study, the authors generated and then used a MEF line deficient in Gphr, encoding a Golgi proton counter ion transporter previously reported to lead to an increased luminal pH in the Golgi apparatus. In line with previous observations, using glycomics and gel-shift assays the study revealed that loss of GPHR in cultured MEFs leads to abnormal glycosylation patterns. Using confocal microscopy and live-cell imaging, the authors also assessed the localization and trafficking of Golgi enzyme reporter constructs and spotted differences in Golgi FRAP and proximal interactions of Golgi enzymes between Gphr-proficient and -deficient MEFs. They then focus specifically on lysosomal membrane proteins that - not unexpectedly - are also affected by altered glycosylation in Gphr-deficient cells. Following drug-induced lysosomal damage/stress, the authors find that Gphr-deficient and -proficient cells display distinct stress response dynamics and come to the conclusion that proper Golgi N-glycosylation is important for lysosomal membrane integrity.

In general, the study is well executed and scientifically sound. It does, however, have a number of imperfections that need to be addressed before I consider this manuscript suitable for publication. Moreover, the scientific implications of the findings presented have a limited scope and would need to be further substantiated to meet the expectations for manuscript suitable for publication in Life Science Alliance.

We thank the reviewer for reviewing our manuscript and for providing encouraging comments on our work. We have carefully addressed the concerns raised and provide point-by-point responses below.

Major points:

The authors present three distinct GPHR KO clones tested by western blot. In some experiments GPHR KO #11 is further characterized. Is this the clone that was used to generate the GPHR (+) and GPHR (-) cells (i.e. stably transfected with plasmid and

empty plasmid, respectively) used in most panels? That should be made clear somewhere. In some cases the authors also use the term "control MEFs". Which cells are meant with that? Parental MEFs with endogenous GPHR or GPHR KO MEFs stably transfected with the empty plasmid?

To begin with, we apologize for the unclear description. In accordance with the reviewer's comment, we have made necessary change to the text and Figures in the revised manuscript. First, we have added a new Figure 1 with the results of our analysis of *Gphr*-deficient and rescued MEFs. Then, to ensure clarity regarding which cells were used in the experiment, we unified the notation of cells into "*Gphr*^{-/-}-GPHR-3F" and "*Gphr*^{-/-}-Vec".

Finally, we have added the following text to the revised manuscript:

"These results support the concept that *Gphr* is restored in *Gphr*^{-/-} #11 MEFs expressing GPHR-3xFlag (*Gphr*^{-/-}-GPHR-3F), but not in MEFs expressing vector control (*GPHR*^{-/-}; Vec) (Hereafter *Gphr*^{-/-}-GPHR-3F and *GPHR*^{-/-}-Vec were used as control and *Gphr*-deficient MEFs, respectively)." (lines 131-134)

Related to that, a limitation of the study is certainly that it is (seemingly) done in one single KO clone and using a Cre-dependent KO system. It needs to be excluded that this is a clonal effect and it would also be of interest if the Golgi phenotype the authors report also holds true in other systems. I feel that the study would benefit from an independently generated CRISPR/Cas9 KO line (possibly a human cell) to further substantiate at least some of the authors' findings.

Based on the reviewer's comment, we performed additional experiments to analyze the clonal effects of cell lines. First, we attempted to establish a *GPHR* KO line using HeLa cells, but this was not successful. This is because there are two *GPHR* genes (*GPR89A* and *GPR89B*) in the human genome, and it is difficult to knock off both simultaneously (Figure for reviewers 1; the data also showed that GPHR protein levels were decreased, but there was no evidence of abnormal glycosylation. Even a slight GPHR expression may be functionally sufficient). Therefore, we established *Gphr*-deficient cells using N2A cells derived from mice. Analysis of *Gphr*-deficient N2A cells showed the same phenotypes as that of MEF cells: elevated lumen pH, abnormal morphology of the Golgi apparatus, abnormal glycosylation, impaired dynamics of glycosyltransferases, and decreased lysosomal membrane stability. These results have been incorporated into the text and Supplementally Figures 2, 5, 6, and 9 of the revised manuscript.

The authors do in fact observe a very substantial glycosylation phenotype (in particular,

when looking at the gel shift assays) in their KO model. They do see differences in regards to the localization/dynamics of glycosyltransferases that might explain this effect. A more straightforward explanation could however also be that Golgi enzymes are somewhat destabilized in a Golgi apparatus with a dysregulated luminal pH and a reduced extent of glycosylation could simply be a consequence of reduced glycosyltransferase levels in the Golgi apparatus. This needs to be tested but cannot be done on ectopically expressed reporter constructs. Instead, the authors should have a close look (e.g. by immunoblotting) at the protein levels of cell-endogenous Golgi enzymes (such as B4galt1 and st6gal1 but possibly also other) in WT vs. GPHR KO cells.

We thank the reviewer for this insightful comment. The results of the additional work suggested by the reviewer have been presented in the Result section and Supplemental Figure 4 of the revised manuscript. According to the results, there were no quantitative differences in the glycosyltransferase levels between both MEFs. (lines 160-164)

The authors observe a substantial glycosylation phenotype and given the profound importance of glycosylation as a modification of proteins and lipids of the eukaryotic pathway this can absolutely be expected to have far-reaching physiological implications for many cellular processes. In the second part of the study the authors then assess the GPHR Kos glycosylation impact specifically in the context of lysosomal membrane proteins. The connection of these two aspects of the story is somewhat "bumpy", though and could possibly be improved for the benefit of an interested reader.

In accordance with the reviewer's comment, we have modified the Abstract and Introduction section of revised manuscript. (lines 38–39 and 87-90)

The authors performed various experiments to assess lysosomal membrane integrity (lysotracker retention, TFEb activation, and Galectin-3 recruitment) following treatment of the cells with the cationic amphiphilic drug terfenadine. I think the wording of the description of the data should be changed as also GPHR-proficient cells (i.e. cells with an intact lysosomal glycocalyx) do display lysosomal damage and are absolutely not protected. The authors can, however, quantify a statistically significant difference in stress response parameters between GPHR-deficient and - proficient cells. But the findings are not adequately described and the authors do not come to a specific enough conclusion whether GPHR does benefit lysosomal membrane integrity or not. The wording should be made clearer here. Also, a dose-response response experiment could be better suited to visualize whether GPHR- cells are more sensitized to

terfenadine-induced lysosomal membrane damage.

In accordance with the reviewer's comment, we have made necessary changes to the text. In the Result section, we have incorporated a sentence stating that *Gphr* deficiency affects lysosomal membrane stability. (lines 315-317). Then, we have revised "protect" to "contribute to stability". (lines 325-326). Similarly, in the Discussion section, we stated that GPHR contributes to lysosomal membrane stability without using the word "protect". (lines 447-449).

Based on reviewer's comment, we treated MEFs with different concentrations and at different time points with Terfenadine and then performed Galectin-3 staining (i.e., lysosomal membrane rupture) to identify the point at which the drug was effective only in *Gphr*-deficient MEFs. Though, we could not find the particular point of Galectin-3 dots exclusively in *Gphr*-deficient MEFs (Figure for reviewers 2) due to incomplete experimental settings (at lower concentrations than those used in this study, no Galectin-3 dots were observed.). Thus, if in the Editor's opinion, it is necessary to remove the data that shows no significant difference (Figure 6) for clearer presentation of the data for the readers, we offer to remove the concerned images and graphs.

The authors address the role of the lysosomal glycocalyx in the context of the GPHR KO model. But these findings obviously are not limited to this model and should similarly hold when Golgi glycosylation is impaired in other ways. To substantiate the generalized importance of complex Golgi N-glycosylation for lysosomal membrane integrity, similar experiments need to be performed in cell culture models that display glycosylation defect due to other molecular defects. This could for instance be possibly done in cells deficient for key N-glycosylation enzymes (e.g. MGAT1 KO) or simply cells treated with N-glycosylation inhibitors such as kifunensine or swainsonine.

We thank the reviewer for this insightful comment. The results of the additional work suggested by the reviewer have been presented in the Result and Discussion sections, and in Supplementally Figure 10 of the revised manuscript. According to the results, glycosylation of lysosomal membrane proteins contributed to stability of lysosomal membrane, protecting from lysosomal damage. (lines 318-326)

Minor points

The KO validation is provided in Suppl. Fig. 1 - but in Fig. 1A data are presented from KO cells rescued with an empty plasmid or a GPHR-3xFlag plasmid. Thus, it is not a comparison of WT and KO cells. That needs to be made clearer in the figure. In fact, a

GPHR western blot (like the ones found in panel C and E) needs to be provided for panel 1A, to make it absolutely transparent to the reader right at the beginning which model is used.

Related to that, the rescue cells do actually express substantially more GPHR than the parental MEFs, but e.g. in terms of glycosylation they do not show any differences to parental cells (which is good). Would an overexpression be actually expected to lower luminal pH further?

We apologize for this ambiguity in the original manuscript. In accordance with the reviewer's comment, we have made necessary changes to the text and Figures in the revised manuscript. First, we have added a new Figure 1 with the results of our analysis of *Gphr*-deficient and rescued MEFs. Then, to establish clarity regarding which cells were used in the experiment, we ensured consistency in used cell notations: "*Gphr*^{-/-}-GPHR-3F" and "*Gphr*^{-/-}-Vec". In addition, we have added the results for measurement of pH of the Golgi apparatus of *Gphr*-deficient MEFs expressing GPHR-3F in Figure 1F. According to the results, luminal pH of the Golgi apparatus in *Gphr*^{-/-}-GPHR-3F was not different when compared with parental MEFs. (lines 125-127)

Re: Complex formation of glycosyltransferases: The wording should be changed when it comes to describing their PLA findings. A PLA signal alone is not sufficient to claim two molecules actually form a stable complex. They are certainly in close proximity, though.

In accordance with reviewer's comment, we have revised "complex formation" to "proximity condition" in the revised manuscript.

*Citing rather old literature that used chemical/pharmacological compounds that alter organellar pH in cell culture models, the authors stress the advantage of KO models as they could more specifically alter pH in specific organelles such as the Golgi apparatus without having a rather broad impact. While this is a valid point, KO models do carry the risk that cells phenotypically adapt and thus could lead to artefactual observations. This risk should be discussed and, given that only one single clonal *Gphr* KO line was used in the study, it would be interesting to what extent the authors observations hold true in an acute depletion setting, e.g. shortly following genome editing in CRISPR/Cas9-targeted *Gphr* KO pools or following siRNA-mediated *Gphr* targeting.*

Based on the reviewer's comment, we performed additional experiments to analyze whether the phenotype of cloned *Gphr*-deficient MEFs is also true in the acute depletion

model. Using adenovirus Cre recombinase expression systems, acute depletion of *Gphr* resulted in a phenotype similar to that of clonal *Gphr*-deficient MEFs, including elevated luminal pH, abnormal morphology of the Golgi apparatus, and abnormal glycosylation. In addition, the *Gphr*-deficient N2A cells exhibited similar results. Elevated luminal pH, abnormal morphology of the Golgi apparatus, and abnormal glycosylation were caused by depletion of *Gphr*. The results of the additional studies are now shown in the Result sections and Supplementally Figure 1, 2 and 5 in revised manuscript. (lines 134-136 and 164-166)

Denomination of glycosyltransferases is inconsistent as sometimes the gene name is used (a useful way of naming of the glycosylation enzymes these days) (e.g. B4galt1) and sometimes the more conventional enzyme naming (e.g. GnTI for Mgat1). This should be done consistently throughout the manuscript. In case of the glycosyltransferases the correct notation for mouse gene names is used. But at the same time the authors always refer to their cell model as GPHR KO - which is not the correct notation of a mouse gene name.

We apologize for the confusion. In accordance with reviewer's comment, we have made the necessary corrections to the text. In the revised manuscript, the gene name of glycosyltransferase used in our study has been standardized to depict mouse gene names. However, the gene names cited in previous studies were based on the species from which they originated. Regarding *GPHR*, we first explained that it is *Gpr89*, subsequently, stated that *Gphr* will be used hereinafter.

Missing data panels and incorrect data panel referencing in figure legends:

Based on the legend, panel D should contain immunostainings of the Golgi apparatus; that panel is missing. The legend lists electron micrographs for panel E, though. The missing data should be included and the main text should be changed to also refer to both panels. In light of this imperfection, the authors should carefully re-check all figure panels and the corresponding legends again to make sure that all data sets are accurately described.

We apologize for these errors. In accordance with the reviewer's comment, we have carefully re-examined all the data sets and made appropriate revisions in the revised manuscript.

There is a similar issue with Fig. 1, in the legend a description of panel B is named as

panel C, panel D mentioned in the legend should be corrected to C and so on.

We apologize for these errors. In accordance with the reviewer's comment, we have carefully re-examined the data sets in revised manuscript.

Electron microscopy: Not mentioned in the main text, the electron micrographs point to a substantially alter Golgi morphology. It would be good to describe this in a bit more detailed manner in the main text and also to label the images to help readers not well acquainted with ultrastructural images spot important differences

In accordance with the reviewer's comment, we have incorporated a sentence into the Results and Discussion sections that addresses the abnormal morphology of the Golgi apparatus. Furthermore, we added arrows and arrowheads to the electron microscope images to better illustrate structure of the Golgi apparatus. (lines 127-129, 134-136 and 414-429).

Preferably, individual data points should be used instead of box-plots.

In accordance with the reviewer's comment, we have changed "box-plot graph" to "bar graph" with individual data points in all figures.

Fig. 5A - lysotracker is misspelled

We apologize for the typographical error. This has been corrected.

Response to Reviewer 2:

We express our appreciation to the reviewer for the insightful comments. Indeed, the comments have helped us to significantly improve the manuscript.

Reviewer #2 (Comments to the Authors (Required)):

This paper reports an examination of the effects of removing from mouse fibroblasts the Golgi membrane protein Golgi pH regulator (GPHR). The Golgi is known to be acidified, especially toward the trans side of the Golgi stack, and previous work has shown that GPHR plays a role in this process and so when it is deleted, Golgi pH is perturbed. Previous work from the authors has shown that in the absence of GPHR, the Golgi processing of protein glycans is perturbed with some modifications becoming less prevalent. This new paper extends this work by focusing on the abundant glycoproteins that are resident in lysosomes and are thought to help protect the lysosomal membrane against the harsh environment of the lysosomal interior. They present convincing evidence that the glycosylation of the various lysosomal associated membrane proteins is substantially affected when GPHR is deleted, along with a concomitant perturbation of Golgi structure and diffusion of Golgi enzymes within the Golgi stack. They go on to show that the membranes of lysosomes are more susceptible to damage in the mutant cells when compared to wild-type. The effects are relatively small, but this is not unexpected given that GPHR loss causes only partial loss of glycan modifying activity, and the effects they do see are carefully quantified and demonstrated with multiple well-established assays.

This paper thus extends previous work on the importance of GPHR and hence Golgi pH for glycosyltransferase activity in the Golgi. It provides valuable evidence that defects in the Golgi can indirectly effect lysosome integrity which is of use for interpreting the effects of gene knockout experiments. There are a couple of relatively minor points that I noticed, but if these are addressed then I would be supportive of publication.

We thank the reviewer for reviewing our manuscript and for providing positive and encouraging comments regarding our work. We have carefully addressed the comments raised by the reviewer and provide our responses below.

A) The authors should note somewhere in the text that the gene name for GPHR is Gpr89 (GPR89 in humans).

In accordance with the reviewer's comment, we have made the necessary change to the text. (lines 61-62)

B) The authors assay glycans using mass-spectrometry (Fig, 1A and Table 1). However, this method can only detect a subset of glycans and will miss those that are very large such as polylectosamine and glycosaminoglycans. These may well be affected, and so the authors should note this limitation to the assay.

We thank the reviewer for this valuable suggestion. As suggested, we have incorporated a sentence into the Discussion section that addresses the limitations of TOF mass analysis. (lines 351-356)

C) The immunofluorescence images that are single channel should be shown as gray-scale with only the merged images in colour.

In accordance with the reviewer's comment, we have converted the single-channel images to gray-scale.

Response to reviewer 3:

We express our appreciation to the reviewers for their insightful comments on our paper. The comments helped us to significantly improve the paper.

Reviewer #3 (Comments to the Authors (Required)):

The manuscript by Sou et al investigates the consequences of altered pH homeostasis in the Golgi on glycan processing and the function of lysosomal glycoproteins. For this they delete the protein leak channel GPHR in mouse fibroblasts and perform cell biological and glycobiological experiments on the Golgi as well as characterisation of lysosomal proteins and lysosomal membrane integrity. The manuscript is well written and clearly laid out. The findings are consistent with previous literature on pH effects on Golgi glycosylation and the functional models of the glycocalyx in the lysosome lumen. The pH regulation from the point of view of GPHR is novel and this is an interesting addition to the glycosylation literature. I would recommend that the authors consider the following revisions before publication.

We thank the reviewer for reviewing our manuscript and for providing positive and encouraging comments. We have carefully addressed the comments raised by the reviewer and provided our responses below.

1. I feel that the evidence presented for pH alteration affecting glycosyltransferase dimerization is not compelling. The PLA assay, as the authors say, has an effective distance of ca. 40 nm. This is a reasonably long distance in the Golgi, could even span the distance between two adjacent cisternae. The PLA signal in WT cells is therefore not very strong evidence for direct interaction of enzymes; a negative control of a trans-Golgi enzyme that does not show a PLA signal would be nice to see. Moreover, given the observed disruption of Golgi architecture in the GPRH- cells, it is possible that the observed loss of PLA signal is due to enzymes in proximity of each other (but not in a complex) in the same or adjacent cisternae being moved further from each other via fragmentation of the organelle.

We appreciate this comment and agree that the representation was not correct. Therefore, we change “complex” to “proximity condition”. In addition, we added the technical control experiment for PLA in Supplementally Figure 7. According to the results, PLA serves as a useful tool to evaluate proximity condition of glycosyltransferases. Furthermore, based on the comment, we have incorporated a sentence in Discussion section to suggest the possibility of loss of proximity condition

of glycosyltransferase caused by abnormal morphology of the Golgi apparatus in *Gphr*-deficient MEFs. (lines 414-429).

2. It would be good to include more information from the literature where altered pH has already been shown to alter glycosylation and enzyme localisation (e.g. for the Golgi ATPase mutations from patients). Given that the experiments described here are using a different pH regulator, this would I feel be complementary to the current study.

According to the reviewer's suggestion, we have added relevant text to the Discussion section. (lines 372-378 and 383-395).

3. It is worth noting that there is a roughly similar amount of non-Golgi staining in figure 4D for the GPHR+ and GPHR- cells shown, suggesting that non-Golgi localisation of GSII staining is not a GPHR- only case. With this in mind, figure 4E showing a different confocal section, which does not seem to contain the Golgi and has a large amount of extra GSII staining not only in GPHR- but also in GPHR+ cells. In GPHR+ cells there is a preinuclear GSII staining cluster that does not look like the Golgi (it is rather punctate) - The authors may want to comment about what compartment this may be?

We agree with the reviewer's opinion about the issues in GS-II staining. Hence, we have replaced the images and included a high-magnification image. The new images demonstrate that GS-II localizes in the Golgi apparatus in both MEFs. Moreover, GS-II exhibits scattered dots that colocalize with lysosomes in addition to the Golgi apparatus in *Gphr*-deficient MEFs.

Minor points:

1. The abbreviation for lysosomal membrane proteins is inconsistent through the manuscript and the abstract.

We apologize for the typographical error. We have corrected this in text at all relevant places. (lines 40 and 47)

2. In table 1 some of the drawn N-glycan structures are not possible - eg hybrid with 6 mannoses and a glycan with 4 mannoses

We thank the reviewer for pointing this out. We have corrected this error in the revised table.

C: Cas9-control
 G1: Cas9-GPHRA-1 + GPHRB-1
 G2: Cas9-GPHRA-1 + GPHRB-2
 G3: Cas9-GPHRA-2 + GPHRB-1
 G4: Cas9-GPHRA-2 + GPHRB-2

Immunoblotting analysis of lysates from GPHR-deficient HeLa cells using the CRIPR-Cas9 system.

1-10 μ M Terfenadine was treated for 1-12 hours.
 Immunofluorescence analysis performed for Galectin-3 and LAMP-1.
 Representative images are shown. Arrows indicate puncta of Galectin-3.

July 15, 2024

RE: Life Science Alliance Manuscript #LSA-2024-02677-TR

Dr. Yu-Shin Sou
Juntendo University School of Medicine
Department of Cell Biology and Neuroscience
Hongo 2-1-1
Bunkyo-ku, Tokyo 113-8421
Japan

Dear Dr. Sou,

Thank you for submitting your revised manuscript entitled "Golgi pH homeostasis stabilizes lysosomal membrane through N-glycosylation of membrane proteins". We would be happy to publish your paper in Life Science Alliance pending final revisions necessary to meet our formatting guidelines.

- please address the remaining points from both Reviewers
- please be sure that the authorship listing and order is correct
- please upload your Table in editable .doc or Excel format
- please add the Twitter handle of your host institute/organization as well as your own or/and one of the authors in our system
- the contributions selected for Yasuo Uchiyama and Masato Koike do not qualify them for authorship. Please either update the contributions in our system and the Author Contributions section of the manuscript or let us know if the authors need to be removed (and added instead to the Acknowledgments section)
- please add a Conflict of Interest statement to your main manuscript text
- please add your main, supplementary figure, and table legends to the main manuscript text after the references section
- we encourage you to revise the figure legends for Figure 2 such that the figure panels are introduced in alphabetical order
- Figure S3 has only one panel. There is no need to mark it with A. Please remove A from both the figure and its legend
- there is a call out for Figure 4G, and this figure doesn't have this panel; please correct
- please add callouts for Figures 5G; S1A-C; S4A-B; S5A-C; S6A-B and S9A-E to your main manuscript text
- please use the [10 author names, et al.] format in your references (i.e. limit the author names to the first 10)

FIGURE CHECKS:

- please add boxes to indicate the areas zoomed-in on for Figure S9, panel D, Gphr/- #25, 0 min

A. FINAL FILES:

- An editable version of the final text (.DOC or .DOCX) is needed for copyediting (no PDFs).
- High-resolution figure, supplementary figure and video files uploaded as individual files: See our detailed guidelines for preparing your production-ready images, <https://www.life-science-alliance.org/authors>
- Summary blurb (enter in submission system): A short text summarizing in a single sentence the study (max. 200 characters)

including spaces). This text is used in conjunction with the titles of papers, hence should be informative and complementary to the title. It should describe the context and significance of the findings for a general readership; it should be written in the present tense and refer to the work in the third person. Author names should not be mentioned.

B. MANUSCRIPT ORGANIZATION AND FORMATTING:

Sincerely,

Reviewer #1 (Comments to the Authors (Required)):

The authors have invested quite some effort - including extensive experimental work and the generation of novel cellular models - to address the reviewers' comments. In doing so, the authors extended and further corroborated the findings reported in the previous version of the manuscript. Specifically, the points I had raised were sufficiently addressed and clarified. Hence, I recommend that the revised version of the manuscript is accepted for publication in LSA - but would suggest to address the following minor points (merely text editing).

Minor points:

Title - I think it would be better to change the title to "Golgi pH homeostasis stabilizes lysosomal membranes through..." or "Golgi pH homeostasis stabilizes the lysosomal membrane through..."

Line 129 - "expressing" instead of "expressed"?

Line 140 - the N2A cells are a good additional model that confirms the authors' observation, but given their murine origin (similar to the MEF cells) and the fact that in humans GPHR is encoded by two homologous genes, this statement should be changed to "in cultured murine cells"

Reviewer #3 (Comments to the Authors (Required)):

Please refer to my original review as well. Here I am only providing an evaluation of the rebuttal.

The authors have provided a large amount of extra data and clarified most points raised by myself, and (in my opinion) by the other reviewers as well.

One point of contention is the interpretation of the PLA results with regards to complex formation between glycosyltransferases.

The authors conceded both to myself and reviewer 1 that PLA can only report proximity, not direct protein complex formation due to the effective distance of the method. They did add some altered text to say proximity in the relevant section. However, they still start the description of the PLA experiments with the rationale that they are using this method to assess complex formation between galactosyltransferase and sialyltransferase. This premise has to be altered, as PLA is not appropriate to assess this question.

All other points in the manuscript are now fine.

Our responses to the reviewers' comments are provided below.

Reviewer #1 (Comments to the Authors (Required)):

Responses to the Comments of Reviewer 1

The authors have invested quite some effort - including extensive experimental work and the generation of novel cellular models - to address the reviewers' comments. In doing so, the authors extended and further corroborated the findings reported in the previous version of the manuscript. Specifically, the points I had raised were sufficiently addressed and clarified. Hence, I recommend that the revised version of the manuscript is accepted for publication in LSA - but would suggest to address the following minor points (merely text editing).

We would like to thank the reviewer for the positive feedback on our manuscript and for the recommendations for publication.

Minor points:

Title - I think it would be better to change the title to "Golgi pH homeostasis stabilizes lysosomal membranes through..." or "Golgi pH homeostasis stabilizes the lysosomal membrane through..."

In accordance with the reviewer's comment, we have made necessary changes.

Line 129 - "expressing" instead of "expressed"?

We thank the reviewer for their comment. We have revised the text accordingly.

Line 140 - the N2A cells are a good additional model that confirms the authors' observation, but given their murine origin (similar to the MEF cells) and the fact that in humans GPHR is encoded by two homologous genes, this statement should be changed to "in cultured murine cells"

We thank the reviewer for their comment. We have revised the text accordingly.

We appreciate the reviewer's constructive and thoughtful comment.

Responses to the Comments of Reviewer 3

Reviewer #3 (Comments to the Authors (Required)):

Please refer to my original review as well. Here I am only providing an evaluation of the rebuttal.

The authors have provided a large amount of extra data and clarified most points raised by myself, and (in my opinion) by the other reviewers as well.

One point of contention is the interpretation of the PLA results with regards to complex formation between glycosyltransferases. The authors conceded both to myself and reviewer 1 that PLA can only report proximity, not direct protein complex formation due to the effective distance of the method. They did add some altered text to say proximity in the relevant section. However, they still start the description of the PLA experiments with the rationale that they are using this method to assess complex formation between galactosyltransferase and sialyltransferase. This premise has to be altered, as PLA is not appropriate to assess this question.

All other points in the manuscript are now fine.

We appreciate the reviewer's thorough assessment and acknowledgment of the additional data and clarification provided. We have addressed the remaining contention concerning the interpretation of the PLA results. Specifically, we refined the description of the PLA experiments to align with the revised terminology. The revised text clarifies that our use of PLA aims to assess the interaction between galactosyltransferase and sialyltransferase rather than to determine direct complex formation (line 198-208). This adjustment ensured the accuracy in presenting the purpose of the PLA experiments. Furthermore, we have referenced a relevant study (Soderberg et al., 2006) in the revised manuscript to support the use of PLA for evaluating glycosyltransferase interactions. Although PLA primarily indicates proximity rather than direct complex formation, it remains a valuable tool for inferring potential interactions.

We acknowledge the reviewer's concern regarding Golgi morphology, which potentially

influences PLA signaling. Notably, a PLA detection distance of 40 nm exceeded the dimensions of the Golgi cistern. We attempted to mitigate this concern but faced limitations in reducing the PLA signal adequately. Additionally, the observed decrease in PLA signaling in GPRH-deficient cells may reflect altered Golgi morphology rather than a direct loss of glycosyltransferase interactions. Nevertheless, even if the reduced PLA signaling stems from morphological abnormalities of the Golgi apparatus, it still suggests a decrease in glycosyltransferase interactions due to the loss of trafficking dynamics of glycosyltransferase.

The reviewer's insights, particularly regarding protein complex interpretation, are valuable and will guide the future development of alternative evaluation systems.

We appreciate the reviewer 's constructive comments.

July 19, 2024

RE: Life Science Alliance Manuscript #LSA-2024-02677-TRR

Dr. Yu-Shin Sou
Juntendo University School of Medicine
Department of Cell Biology and Neuroscience
Hongo 2-1-1
Bunkyo-ku, Tokyo 113-8421
Japan

Dear Dr. Sou,

Thank you for submitting your Research Article entitled "Golgi pH homeostasis stabilizes the lysosomal membrane through N-glycosylation of membrane proteins". It is a pleasure to let you know that your manuscript is now accepted for publication in Life Science Alliance. Congratulations on this interesting work.

DISTRIBUTION OF MATERIALS:

Again, congratulations on a very nice paper. I hope you found the review process to be constructive and are pleased with how the manuscript was handled editorially. We look forward to future exciting submissions from your lab.

Sincerely,
